


# Integrated sea storm management strategy: the 29 October 2018 event in the Adriatic Sea

Christian Ferrarin[1], Andrea Valentini[2], Martin Vodopivec[3], Dijana Klaric[4], Giovanni Massaro[5], Marco Bajo[1], Francesca De Pascalis[1], Amedeo Fadini[1], Michol Ghezzo[1], Stefano Menegon[1], Lidia Bressan[2], Silvia Unguendoli[2], Anja Fettich[3], Jure Jerman[3], Matjaž Licer[6], Lidija Fustar[4], Alvise Papa[5], and Enrico Carraro[7]

[1]CNR - National Research Council of Italy, ISMAR - Marine Sciences Institute, Venice, Italy
[2]Arpae-SIMC - Agency for Prevention, Environment and Energy of Emilia-Romagna, Hydro-Meteo-Climate Service, Bologna, Italy
[3]Slovenian Environment Agency, Ljubljana, Slovenia
[4]Meteorological and Hydrological Service of Croatia, Zagreb, Croatia
[5]Tide Forecast and Early Warning Center, City of Venice, Venice, Italy
[6]National Institute of Biology, Ljubljana, Slovenia
[7]City of Venice - European Policies, Venice, Italy

**Correspondence:** Christian Ferrarin (c.ferrarin@ismar.cnr.it)

**Abstract.** Addressing coastal risks related to sea storms requires an integrative approach which combines monitoring stations, forecasting models, early warning systems and coastal management and planning. Such great effort is sometimes possible only through transnational cooperation, which becomes thus vital to face effectively and promptly these marine events which are responsible for several damages impacting on the environment and citizens' life. Here we present a shared and interopera-
ble system to allow a better exchange and elaboration of information related to sea storms among countries. The proposed Integrated Web System (IWS) is a combination of a common data system for sharing ocean observations and forecasts, a multi-model ensemble system, a geoportal and interactive geo-visualization tools to make results available to the general public. Multi-model ensemble mean and spread for sea level height and wave characteristics are used to describe three different sea condition scenarios. IWS is designed to provide sea state information required for issuing coastal risk alerts over the analysed
region, as well as for being easily integrated into existing local early warning systems. This study describes the application of the developed system to the exceptional storm event of 29[th] of October 2018, that caused severe flooding and damages to coastal infrastructures in the Adriatic Sea. The forecasted ensemble products were successfully compared with in situ observations. The hazards estimated by integrating IWS results in existing early warning systems were confirmed by documented storm impacts along the coast of Slovenia, Emilia-Romagna and the City of Venice. For the investigated event, the most severe
simulated scenario resulted to provide a realistic and conservative estimation of the peak storm conditions to be used in coastal risk management.



## 1 Introduction

Sea storms represent the main threat in coastal areas. In fact, they directly impact on the citizens' quality of life (especially in urban areas where part of the inhabited areas is seldom covered by water), they create damages to the important cultural heritage exposed to these phenomena, and they affect businesses too (aquaculture, fisheries, tourism, beach facilities) and environment
at large (coastal erosion, floods) (Chaumillon et al., 2017; Reimann et al., 2018; Vousdoukas et al., 2018a). The potential future effects of global climate change emphasise the need for strategies based on an anticipatory approach particularly in coastal areas at immediate and high risk (Hinkel et al., 2014; Vousdoukas et al., 2018b). This is particularly true for coastal wetlands if enough additional accommodation space will not be created through careful nature-based adaptation solutions to coastal management (Schuerch et al., 2018).

Coastal flooding is induced by extreme sea levels, determined by the increase in sea level caused by strong winds and low atmospheric pressure (storm surge), often in combination with high tides (Muis et al., 2016). Under such extreme meteorological conditions, the coast could be also vulnerable by stormy waves with potential damages to infrastructures and erosion. Moreover, when waves reach the coast they interact with the bathymetry and drive an additional increase in water levels through wave setup (Roland et al., 2009) and they travel up and down the beach before being reflected seaward. The maximum vertical
excursion of wave uprush on a beach or structure above the still water level is called the wave runup (Sorensen, 1997).

Coastal flooding, erosion, impacts on ecosystems, damages to infrastructures and productive activities can worsen if combined with the absence of adequate early warning systems, coordinated strategies, intervention procedures, coastal management and planning with significant related economic costs (Hinkel et al., 2014; Prahl et al., 2018). The difficulty of reacting promptly to extreme events is also connected to the lack of shared data and know-how. Recognizing the importance
of information sharing for disaster risk reduction, human safety and well-being, the World Meteorological Organization (WMO, https://public.wmo.int/) promotes the standardization and exchange of observations since 1873. Similarly, the Permanent Service for Mean Sea Level (PSMSL, http://www.psmsl.org/) and the Global Sea Level Observing System (GLOSS, http://www.ioc-sealevelmonitoring.org/) are responsible for the collection, publication, analysis and interpretation of sea level data from the global network of tide gauges. In the same direction, at European level, the Copernicus Marine Environmental
Monitoring Service (CMEMS, http://marine.copernicus.eu/), the European Marine Observation and Data Network (EMODnet, http://www.emodnet.eu/) and the European Global Ocean Observing System (http://eurogoos.eu/) aim at sharing information from both satellite and in situ observations, as well as state-of-the-art analyses and daily forecasts, which offer an unprecedented capability to observe, understand and anticipate marine environment events.

Despite such international effort on data sharing, as weather, climate and ocean know no national boundaries, the insufficient
level of cooperation among neighbouring countries is often a cause of ineffective actions at local level and missed opportunities to collaborate with other actors to increase overall preparedness to sea storms (Chaumillon et al., 2017).

The problem of sea storms is particularly relevant for the Adriatic Sea, where extreme sea levels are higher than in other parts of the Mediterranean basin (Marcos et al., 2009) and several coastal cultural World Heritage sites (http://whc.unesco.org/) at risk from coastal flooding and erosion are located (Prahl et al., 2018; Reimann et al., 2018). This study presents the management





approach for sea storm hazard initiated as part of the I-STORMS (Integrated Sea sTORm Management Strategies) project for the coastline of the Adriatic-Ionian macro-region (https://istorms.adrioninterreg.eu/). This manuscript describes a joint strategy to safeguard the coast from sea storm emergencies by sharing knowledge, data and forecasts among involved countries and improving their capacities in terms of early warning and management procedures. This study focuses on the recent exceptional

storm event of 29th of October 2018, which is here taken as a pilot study for applying and testing the developed approach.

## 1.1 Study area

The Adriatic and Ionian seas are part of the Mediterranean Sea positioned between the eastern coastline of Italy, countries of the Balkan Peninsula (from Slovenia, south through Croatia, Bosnia-Herzegovina, Montenegro and to Albania) and Greece. The Adriatic Sea is an 800-km-long, 150-km-wide elongated semi-enclosed basin interacting with the Ionian Sea through the

10 Otranto Strait in the southern part (Fig. 1). The shallow northern Adriatic Sea is the Mediterranean sub-basin where storm surges reach higher values (Marcos et al., 2009), mainly triggered by strong south-easterly moist and warm wind, called Sirocco. For this reason, in this area storm surges and waves have been deeply investigated in the past (Orlić et al., 1992; Bajo and Umgiesser, 2010; Cavaleri et al., 2010; Lionello et al., 2012; Medugorac et al., 2015; Ferrarin et al., 2017; Pomaro et al., 2017; Vilibić et al., 2017; Bajo et al., 2019; Ferrarin et al., 2019). Tidal dynamics are particularly evident in the northern

Adriatic Sea, where the most energetic tidal constituents, the semi-diurnal $M_2$ and the diurnal $K_1$, reach amplitudes of 27 and 18 cm, respectively (Ferrarin et al., 2017).

The weather in the Adriatic area is strongly influenced by local orography and small-scale processes (Pasarić et al., 2009). The use of high-resolution meteorological models is essential to capture the temporal and spatial inhomogeneity of north-easterly Bora winds, characterised by topographically controlled high-speed wind jets along the eastern shore (Signell et al.,

2005; Davolio et al., 2015). The same holds for Sirocco: global and regional numerical models have been shown to consistently underestimate its speed due to the fact that orography, and hence the channelling of the air flow, is not well represented at typical model resolution (Cavaleri and Bertotti, 2004). Long term analyses of general wind conditions over the Adriatic basin further indicate a trend of reduction of the intensity of wind events - mostly due to Bora (Pirazzoli and Tomasin, 2003) and a general increase in terms of frequency, mostly associated to the increasing storminess of Scirocco (Pomaro et al., 2017).

The eastern and western coasts of the Adriatic Sea greatly differ in appearance and are therefore differently impacted by sea storms. The western coast is largely sedimentary with mild sloping and sandy beaches, while the eastern coast is composed of many islands and headlands and is generally high and rocky. Due to its alluvial origin, natural subsidence occurs in the northwestern Adriatic Sea because of compaction of fine-grained deposits (Carbognin and Tosi, 2002), that is worsened by the human exploitation of underground water and gas in some areas. Several shallow coastal transitional water bodies are present

along the Italian coastline, the main of which are the Marano-Grado Lagoon, the Venice Lagoon, the system of lagoons of the Po Delta, the Lesina Lagoon and the Varano Lagoon (Umgiesser et al., 2014).

Extreme sea levels cause the flooding of several coastal cities on both sides of the Adriatic Sea (Lionello et al., 2012; Medugorac et al., 2015), especially when the storm is associated with spring tides (Bajo et al., 2017). Part of the western coast is below sea level, and therefore it is also very vulnerable to such hazards (Lionello et al., 2012). These coastal zones are also



strongly impacted by north-easterly storms with severe morphological impacts on natural sectors and damage to structures along urbanised zones (Armaroli et al., 2012; Harley et al., 2016). Conversely, recurrent meteotsunami events occur on the eastern side of the Adriatic Sea, and particularly the Croatian coast and islands, causing flooding and damage in some harbours (Orlić, 2015). According to Rizzi et al. (2017) and Satta et al. (2017), the northern Adriatic coastline, due to its low elevation, will be one of the regions in the Mediterranean area most exposed - in terms of coastal risk for flooding and erosion - to future climate change.

## 2   Material and methods

In order to address the territorial challenges related to sea storms effect on the coastal areas, we developed a shared and interoperable system (Integrated Web System - IWS) to allow a better exchange of information at a basin scale. Therefore, available resources can be accessed simultaneously in an aggregated and standard way. IWS was designed to specifically store, visualize and share the following category of geospatial and informative contents:

a. historical and real-time (or near real-time) time series of observations from fixed-point sensor networks;

b. outputs from existing operational forecast models;

c. localization and description of coastal sea storm events that have damaged the environment, social-cultural and economic assets;

d. bi-dimensional geospatial layers to provide georeferenced representations of the study area. Such layers are organized in thematic categories (e.g. terrestrial and maritime boundaries, ports, shorelines, morphology and bathymetry, cultural heritage, coastal defence work);

e. datasets, model outputs and time-series metadata to improve discoverability and proper re-use of the shared resources.

All information on coastal disaster due to sea storm events (historical and more recent) are organized and mapped in geospatial layers which constitute the Sea Storms Atlas. That series can be used to draw the map of risk characterisation of the coast with the aim of identifying the most vulnerable areas and supporting the planning of coastal area use and development (Depellegrin et al., 2017).

The IWS implementation is based on Free and Open Source Software and the architecture design follows a resource-centred and service-oriented approach as described in Yang et al. (2007) and Longueville (2010). Following the so-called Service-Oriented Geoportal Architecture, the IWS includes three main layers:

– the resource layer corresponds to the physical storage of the structured information in databases or files;

– the access layer includes all code and software designed to provide access to the resources in the appropriate format;

– the Graphical User Interface (GUI) is the client-side component of the Geoportal architecture; the role of GUIs is not limited to the rendering of a given set of resources but also includes the aggregation of relevant resources through



lightweight and loosely coupled JavaScript code. In other words, the GUI is not only a presentation layer but also creates a mash-up of relevant resources.

IWS overall architecture is described in Fig. 2. Furthermore, the schema highlights the user typologies served by the IWS and the interactions/connections with the partner's nodes and with external portals. IWS is structured into six main components:

1. the *Resource Layer* for storing the datasets, metadata, resources and all the necessary information. It consists of a combination of different storage solutions in order to support the several and heterogeneous data models and formats shared, and all the information needed to achieve a fully operational infrastructure (e.g. metadata, catalogue information, user accounts and profiles);

2. the *Data importer* for data ingestion, harmonization, preparation and deposit the datasets in the storage facilities of the
10 Resource Layer. For this purpose, we implemented the use of data servers (e.g. THREDDS, Hyrax) with the advantages that such web systems are open-source and they already implement services like DAP (Data Access Protocol), WCS (Web Coverage Service), WMS (Web Map Service), SOS (Sensor Observation Service);

3. the *Transnational Multi-model Ensemble System (TMES)* for collecting and combining the results from existing operational forecast systems (described in section 2.2);

4. the *Task Manager middleware* for orchestrating the communication with IWS components (e.g. Data importer, TMES) in order to launch the process (e.g. download the data from the partners' node), monitor the execution status, and concatenate multiple tasks in a single processing pipeline. The Task Manager middleware supports a time-based job scheduler, synchronous-asynchronous task queue system and a message broker system;

5. the *Common Data Sharing System* (CDSS) (Access Layer) for publishing the API and the web services to interact (e.g.
search, visualize, download, manage) with the informative resources through standardized interfaces (e.g OGC-Web service, web API);

6. the *Geoportal* (Graphical User Interface) for implementing the end user interfaces and tools to search, visualize, explore and analyse informative resources. The Map Viewer and Composer is an interactive and dedicated GUI for creating, managing and sharing multi-layered maps and for navigating and querying them.

## 2.1 The monitoring networks

A joint asset which could be exploited through fruitful cooperation is the presence in the whole Adriatic-Ionian coastal territories of large networks of sensors and stations. In the Adriatic region, we mapped 36 tide gauges (9 inside the Venice Lagoon) and 9 wave stations, with the highest concentration in the northern Adriatic Sea. The location of all reported monitoring stations is illustrated in Fig. 1, and their general characteristics are summarized in Table 1 and Table 2, for sea level and wave
respectively. The stations' lists are not exhaustive since there are other monitoring stations active in the area, the data of which were not available at the time of writing this document.





In several cases, the stations are also equipped with sensors for monitoring meteorological (wind speed and direction, sea surface pressure, air temperature, relative humidity and precipitation) or oceanographic parameters (seawater temperature, salinity, current speed and direction).

## 2.2 The forecasting systems

A multi-model ensemble was developed to combine the outcomes of existing ocean and wave forecasting systems, helping in improving the forecast accuracy and reliability on one hand and by adding indications on the forecast uncertainty on the other hand. The error of multi-model ensemble products should be on average lowest compared to those of the ensemble members (Golbeck et al., 2015). According to Di Liberto et al. (2011), operational forecast benefits from the combination of different ocean models by considering different physical parameterization, numerical schemes, model resolution and forcing.

Several operational ocean forecasting models are currently available for the Adriatic-Ionian region. Here we combined 17 forecasting systems, with 10 predicting sea level height (either storm surge or total water level) and 9 predicting wave characteristics. The general characteristics of the forecasting systems are summarized in Table 3 and Table 4, respectively for sea level and wave. We would like to point out that there are other operational systems active in the area (e.g. the pan-European Storm Surge Forecasting System, Fernández-Montblanc et al., 2019), the results of which were not available at the time of
writing this document.

The different operational models are forced at the surface boundary by several meteorological models (ECMWF, BOLAM, MOLOCH, COSMO, WFR and ALADIN) with horizontal resolution ranging from 16 to 1.4 km. The length of the ocean forecast is mostly related to the length of the meteorological forecast and varies from 1.5 to 10 days. There is a large variability in the model's set-up in terms of spatial resolution, temporal frequency, spatial domain (Mediterranean Sea, Adriatic Sea,
northern Adriatic Sea), grid arrangement (e.g. structured or unstructured) and data format (NetCDF, GRIB). Only two of the considered systems (Kassandra and Adriac) account for the current-wave coupling and two forecasting systems perform data assimilation of tide gauge observations in the operational chain (SIMMb and SIMMe).

TMES is implemented as an internal processing engine which interacts directly with the Resource Layer to access the datasets (e.g. time series and forecasts) and to deposit the processing results (e.g. ensemble model result, report, statistics).
Such outputs are available to the end-users and external portal through the Common Data Sharing System and the Geoportal web interfaces.

All numerical model results are interpolated, through a distance-weighted average remapping of the nearest neighbours, on a common regular lat-lon grid covering the Adriatic-Ionian macro-region with a resolution of 0.02 deg. For coastal flooding hazard purpose, the total sea level height must be forecasted. Therefore, the astronomical tidal level values obtained by a
specific SHYFEM application over the Mediterranean Sea (Ferrarin et al., 2018) are added to the residual sea level simulated by the operational systems not accounting for the tide (SHYMED, ISSOS, SIMMb, SIMMe and MFS). The so obtained sea level height simulated by the different models are all referred to the geoid. The spread among the operational simulations is expected to represent a measure of the uncertainty of prediction and should be linked to the forecast error, so that cases with the largest spread are those with the highest uncertainty and where a large error of the ensemble mean (and also of the





deterministic forecast) is more likely (Mel and Lionello, 2014). TMES produces results in terms of the ensemble mean and standard deviation, accounted for a measure of the forecast uncertainty (Flowerdew et al., 2010).

## 2.3 Hazard assessment and early warning systems

The vulnerability to sea storms of a particular segment of coast depends on a wide number of variables, not only related to
5 the magnitude of the storm but including the land characteristics and the social and economic activities that distinguish that area. In order to draw the hazard map showing the region affected by stormy conditions along the Adriatic region, with the aim of identifying most vulnerable areas to a forecasted storm event, the coast is subdivided into segments of variable length in function of morphology, human settlements and administrative boundaries. The coastal assessment units were selected according to the Mediterranean Coastal Database (MCD) developed by Wolff et al. (2018). For each of these units, the database
provides information on the characteristics of the natural and socio-economic subsystems, such as vertical land movement, coastal slope, coastal material and number of people exposed to sea-level rise and to extreme sea levels. Furthermore, the level of vulnerability to sea storm events are complemented with the results of consultation among different public and private socio-economic actors having their main activity on the coast or at the marine level which is affected from the sea storms or bearing responsibilities for informing the citizens.

At each location, three sea condition scenarios are computed considering mean and standard deviation of predicted sea level and wave ensembles:

- MIN: Ens. Mean – Ens. St.Dev

- MEAN: Ens. Mean

- MAX: Ens. Mean + Ens. St.Dev.

Several methodologies have been developed and applied at the basin and local scales for estimating hazard maps for coastal flooding (Hinkel et al., 2014; Vousdoukas et al., 2016; Wolff et al., 2016; Rizzi et al., 2017; Armaroli and Duo, 2018). Over the whole Adriatic-Ionian coastal region, the hazard assessment to sea storm is computed considering the total water level (TWL) obtained combining the sea level height, wave setup and wave runup according to the Stockdon's formula ($R_2$, the 2% exceedance level of runup maxima; Stockdon et al., 2006), using nearshore forecasts provided by the TMES for each coastal
unit. It must be pointed out that the widely used Stockdon's formula - developed for sandy beaches - could underestimate and overestimate wave runup on gravel beaches (Poate et al., 2016) and rocky cliffs (Dodet et al., 2018).

It is well known that the estimation of the total water level is strongly influenced by the local coast typology and morphology and the MCD segments are sometimes too coarse to represent complex morphologies, especially in confined coastal systems (lagoons) and along the eastern rocky coast. Therefore, in order to provide more reliable and resoluted hazard assessment at
30 a very fine coastal scale, the IWS has been designed to be easily integrated into existing early warning systems, developed in areas were a deep knowledge of the coastal dynamics and high-resolution datasets (topography and bathymetry) are available. In this study, we present three existing local forecasting and early warning systems operative in the Adriatic Sea (Slovenia, Emilia-Romagna region and the City of Venice) to which IWS provides the information required for issuing coastal risk alerts.



### 2.3.1 Slovenia

TMES forecasts can be used directly by regional authorities for assessing the hazard of a particular segment of the coast to the storm event according to predefined thresholds. As an example, we report here the IWS based hazard estimates for the Slovenian coast, which is predominantly rocky and steep (flysch cliffs), and therefore well protected from flooding during

storm surges. Important exceptions are the salt pans (Sečovlje and Strunjan) and urban areas such as Piran, Koper and Izola where lower parts get flooded up to 17 times per year (data for the 1963-2003 period; Kolega, 2006), with consequent damage to private properties and cultural heritage. The Slovenian Environment Agency issues a warning when the predicted sea level at Koper exceeds the yellow alert level which is set at 300 cm (above local datum). This is the value that marks the flooding of the lowest coastal urban areas. Orange and red alert levels are set to 330 and 350 cm, respectively.

### 2.3.2 Emilia-Romagna

In addition to the evaluation of thresholds for identifying critical storm conditions at sea (Armaroli et al., 2012), since December 2012, the Emilia-Romagna region (northern Italy) daily provides three-day forecast of coastal storm hazard at eight key sites along the coast, where several past sea storms have induced significant morphological change and damages. The Emilia-Romagna coastline is particularly vulnerable to sea storms due to its low-lying nature and high coastal urbanization (Armaroli

and Duo, 2018). During major storm events, the water levels often exceed those of the dune crest and building foundations (Harley et al., 2016). The existing coastal Early Warning System (Harley et al., 2016) is based on the 1D cross-shore implementation of the XBeach morphodynamic model (Roelvink et al., 2009), a 2DH (depth-averaged) cross-shore process-based model that solves intra-wave flow and surface elevation variations for waves in intermediate and shallow water depths. The XBeach model is used to forecast wave runup and total water level during storm events. For each key site, IWS provides to

the XBeach model the sea level and wave characteristics for the three above-mentioned sea condition scenarios. Hence, the developed methodology allows converting the forecast uncertainty on nearshore sea conditions into a coastal flooding hazard range of predictions. Coastal hazard is here estimated in terms of two storm impact indicators:

- Safe Corridor Width (SCW), a measure of the amount of dry beach available between the dune foot and waterline for safe passage by beach users;

- Building Waterline Distance (BWD), a measure of the amount of dry beach available between the seaward edge of a building and the model-derived waterline.

### 2.3.3 City of Venice

The City of Venice is located in the centre of a shallow lagoon and is composed of more than a hundred islands linked by bridges. The elevation of these islands is extremely low, subjecting them to flooding during storm tides (resulting from

the combination of storm surge and the astronomical tide), which in turn threatens the unique cultural heritage of this city and affects its everyday life, causing among all: difficulties in transport, the practicability of roads and internal channels,





emergency procedure response, commercial activities. In the city of Venice, a bulletin of forecasted sea level up to 3 days is emitted three times per day (at 9 am, 1 pm, 5 pm) by the Tide Forecast and Early Warning Center (CPSM). The forecast is based on a combination of statistical and deterministic models, as well as an evaluation of the synoptic meteorological conditions (https://www.comune.venezia.it/it/content/centro-previsioni-e-segnalazioni-maree).

Since Venice is protected from the sea by barrier islands (separated by three inlets), storm waves do not affect significantly - through setup and runup - the sea level height inside the lagoon (Roland et al., 2009). While propagating from the sea to the lagoon through the inlets, the tidal signal is deformed, either damped or amplified, according to a relationship between local flow resistance and inertia and the characteristics of the incoming open sea signal (Ferrarin et al., 2015). For those reasons, sea level height forecasts are used instead of TWL predictions in the operational system. To propagate the sea level from the inlets
to the inner lagoon, nearshore TMES values of sea level height - for each of the above-mentioned three sea condition scenarios - are referred to the local sea level reference datum (Punta della Salute) and used as open sea boundary conditions in the SHYFEM finite element hydrodynamic model of the Venice Lagoon (Bajo and Umgiesser, 2010; Cavaleri et al., 2019). Such model adequately reproduces the complex geometry and bathymetry of the Venice Lagoon using an unstructured numerical mesh composed of triangular elements of variable form and size (down to a few meters in the tidal channels). Flooding maps
of the city floor are produced by imposing the sea level height predicted at Punta della Salute (at intervals of 10 cm) to a centimetre accurate digital terrain model of the city (http://www.ramses.it/).

The Municipality plan of procedures in case of high and low tide (City of Venice, 2002) defines the actions the several stakeholders (civil protection, public security and rescue forces, transport companies, public services) adopt in case of risk for flooding, with respect to the specific forecasted sea levels. Depending on the forecasted sea level, particular categories
of stakeholders are informed by CPSM and elevated wooden walkways are installed in areas of the city that are prone to flooding. The communication channels for the warning includes a website, messages (SMS, social network), e-mails, phone calls, acoustic signals, direct information (door to door). Moreover, an operating room with forecasters is functioning 24 hours a day at CPSM during the most severe storm tide event.

## 3   The 29 October 2018 event

### 25  3.1   Storm description

On 29 October 2018, an exceptional storm hit the central and northern part of Italy with very strong south-easterly winds (called Sirocco) over the Adriatic Sea. The basic meteorological situation of the 2018 storm was similar to the 1966 and 1979 ones, although with a weaker pressure gradient over the Adriatic area (Cavaleri et al., 2019). The weather condition was governed by a semi-stationary upper level trough which was positioned over West Mediterranean on 28[th] of October and was only slowly
moving north-eastward on 29[th] and 30[th] (Fig. 3). The upper level southerly flow on the East side of the trough was very intense due to strong pressure gradients throughout the whole period of the event. The occurrence of the upper level through resulted in a formation of a very intense low level low-pressure system over the Alps and Central Mediterranean which was the most prominent surface feature of the event.



The air mass over the Italian peninsula and Adriatic was very unstable on 28$^{th}$ and 29$^{th}$ of October due to meridional flow which was bringing moist and warm air from North Africa and Central Mediterranean. In this sense, it was a typical Autumn situation when the amount of precipitation can be extreme, especially on the windward side of orographic barriers. The flow at the surface was further intensified by extreme convection over the Apennines and the Alps. The amount of precipitation in northern Italy and wind storms over the Alps and northern Adriatic were rather extreme and not often observed in such intensity.

It is worth mentioning that Sirocco wind started already on 26$^{th}$ at the most of Adriatic and lasted for almost four days without interruption with the strongest wind in North Adriatic on 29$^{th}$ afternoon, just before the passage of the cold front. Most of this time, the Sirocco was well developed over the entire Adriatic basin and even further south in the Ionian Sea. This meant that the fetch was exceptionally long for the locations in the northern Adriatic Sea.

Consequently, sea level raised in the northern end of the Adriatic Sea - the area most affected by flooding - reaching peak values around 13 UTC in Venice, Koper and Rovinj (Fig. 4a). Exceptional sea levels were reached also along the Emilia-Romagna region with values higher than 1 m for about 5 hours, as measured at Porto Garibaldi. It has to be noted that these maximum values were not registered during the storm peak (happened around 19 UTC in this location) due to an out-of-phase with the astronomical tide. A secondary maximum was recoded in Koper and Rovinj just after the cold front passed and when the wind and waves were decreasing but the tide was rising. Along the central and southern Croatian coast, sea level resulted to be marginally affected by storm surge, even if weak meteotsunamis were observed during the frontal passage late in the evening on the 29$^{th}$ October.

The very long wind fetch allowed waves to develop over the whole basin. Available wave monitoring stations recorded values of significant wave height (the average height of the highest one-third of all waves measuredr; SWH) up to 6 m at the Piattaforma Acqua Alta (PTF), 15 km offshore the Venetian littoral, and up to 4.7 m (8 m of maximum wave height) near the city of Rovinj (Fig. 4b). Along the north-western italian coastline, due to its mild slope, wave breaking occurs and the measured SWH reaches values of about 2 m during the storm peak (Nausica and Senigallia monitoring stations). On the Trieste Gulf, the highest waves occurred 6 hours later (Zarja wave buoy), probably due to the eastward shift of the wind induced by the passage of the cold front. In the south-eastern Adriatic Sea, high wind and wave values were recorded even before the cold front on 28$^{th}$ October. The highest waves recorded in Dubrovnik reached values of about 5 and 9 m for significant and maximum wave height, respectively. Rough sea conditions (SWH > 2.5 m) lasted for 57 hours while the very rough sea state (SWH > 4 m) was recorded for 9.5 hours. According to long-term time series of observations, the 29 October 2018 event reached the records of the second highest sea state ever measured all along the Adriatic coast.

## 3.2 Storm predictability

Here we present the results of the forecasting system at hourly time step and for the day of the event only. However, as described by Cavaleri et al. (2019), up to five (six for the surge) days earlier there were indications of a severe event. Fig. 5 shows the TMES results in terms of the ensemble mean and standard deviation for both the sea level height (panels a and b) and the significant wave height (panels c and d). Storm surge during the 29 October event affected mostly the northern Adriatic Sea




(Fig. 5a), while severe sea condition occurred over most of the Adriatic Sea with the higher waves impacting the Croatian coast (Fig. 5c). The ensemble operational system provides also the estimation of the uncertainty associated with the forecast of this event. Uncertainty is generally higher were the sea level and the waves reach the highest values (Fig. 5b and d). The ensemble standard deviation reached maximum values of about 30 cm for the sea level and 1.5 m for the significant wave height.

Model forecasts could be extracted at any location in the domain to provide a clear representation of sea storm evolution. As an example, we reported in Fig. 6 the values extracted at PTF, Rovinj and Dubrovnik (see Fig. 1 for their location). The comparison with the observations highlights the good performance of the ensemble methodology in reproducing such a strong event. The ensemble mean time-series are smoother than the observations and slightly underestimate the maximum sea level in the northern Adriatic Sea (Figs 5a and 6b), as well as the peak wave height at 20 UTC (5 m of forecasted significant wave

height with respect to almost 6 m of observed at PTF; Fig. 6c). However, the observed values are - almost always - within the TMES spread (i.e. the standard deviation of the ensemble members). It is worth noting that the forecast uncertainty increases with the forecast lead time and with the severity of the storm, the maximum of which was reached in the northern Adriatic Sea between 19 and 20 UTC. In the southern Adriatic Sea (Fig. 6d), the ensemble mean well reproduces the observed significant wave height, which remained between 3 and 5 m for the whole day. For this specific location the spread of the ensemble

assumed values between 0.7 and 1.1 m on 29th October.

### 3.3    Storm hazard and impact assessment on the coast

In order to assess the storm hazard and impact at a basin scale, the results of the multi-model ensemble - in terms of sea level and wave conditions - were processed for each coastal assessment unit of the investigated area. Considering the general underestimation of the ensemble means during the peak of the storm, we will focus our storm hazard analysis on the MEAN and

MAX sea condition scenarios. The total water levels forecasted for the 29 October 2018 event (at 19 UTC) are reported in Fig. 7 for scenarios MEAN and MAX. As for sea level height results (Fig. 5a), the maximum values of TWL are found in the North Adriatic along the Veneto and Friuli Venezia Giulia coasts. Indeed, during the 29 October storm, several coastal lowlands in the northern Adriatic were flooded. At these locations, characterised by gently sloping sandy beaches, the estimated 2% exceedance level of wave runup maxima ($R_2$) reached values of 0.7 m in the MAX sea condition scenario, accounting therefore

for about 50% of the total water level. TWL differences between the MEAN and MAX scenarios reached there the maximum values of about 0.4 m, that is higher than the standard deviation of the multi-model ensemble for the sea level height.

The combination of the sea level height and the wave setup/runup generated high values of the total water level (TWL > 1.5 m, with $R_2$ > 1 m) also along the Istria peninsula, south of Dubrovnik and close to Ancona. Along the Istrian coast, the extremely high waves and the high sea levels caused widespread damages to the coastal infrastructure (Opatija and Zadar).

Moreover, because of the rough sea conditions, there was a disruption of the maritime traffic during the 27-30 October and the ferry service cancelled almost all the scheduled sailings on 29th October, so most of the Croatian islands were cut off from the mainland for more than a day. As stated in section 2.3, previous studies demonstrated that the wave runup estimation increases with the slope of the structure. Therefore, the high wave runup values found at some coastal segments are due to the severe wave conditions, but also to the fact that they are characterized by steep rocky coast (slope > 0.15). On such reflective conditions,





the use of an alongshore-averaged beach slope in practical applications of the runup parameterization may result in large runup error (Stockdon et al., 2006). Moreover, according to Dodet et al. (2018), wave runup could be overestimated at locations with rocky cliffs (e.g. the coast south of Ancona, the Croatian coast and islands), which act on the wave transformation by increasing dissipation and/or shifting offshore the breaking point.

In the following paragraphs we describe the results of the application of the multi-model ensemble to the existing early warning systems and investigate into details the storm hazard and impact at the three selected locations.

Due to its northward orientation, the Slovenian coast is relatively well protected from the waves generated by southerly winds, as in the case of the 29 October 2018 storm. Indeed, over there and for this event the wave contribution to the total water level is negligible. According to the 10-min observation data presented in Fig. 8, the sea level in Koper reached peak

values well above the orange alert level (343 cm at 12:50 UTC and 341 cm at 23:20 UTC) and lasted for almost 10 hours above the yellow alert level. As a consequence, the sea flooded Punta, Prešernovo nabrežje (Prešeren Seafront), Cankarjevo nabrežje (Cankar Seafront), the red pier and Tartinijev trg (Tartini square) in Piran; Veliki trg (Grand Square), Sončno nabrežje (Sunny Seafront) and parts of Dantejeva ulica (Dante street) in Izola. During the storm, the firemen set up anti-flooding barrages at many locations. As shown in Fig. 8, the MEAN scenario predicted the exceeding of the yellow flooding alert level but

underestimated the observed peak values. A more realistic - even if slightly overestimated - prediction of the flooding hazard for the Slovenian coast is provided by the MAX scenario.

Along Emilia Romagna region, several coastal sites were affected by flooding and erosion during the 29 October 2018 sea storm, due to the combination of high sea level and energetic wave conditions. The documented coastal impacts are reported in Fig. 9b and include erosion of the beach and of the winter dunes, coastal flooding and damage to coastal infrastructures

and defence structures. Damages and impacts were notified especially for the northern part of the region, while along the southern coastal area between Cesenatico and Riccione real impact data are not available. The hazard index computed for this event using the XBeach model forced with the three (MEAN, MIN and MAX) conditions, reveals that the most severe sea condition scenario (MAX scenario) provides an exceeding of the predefined alert thresholds indicating a high level of coastal risk. An example of the Safe Corridor Width (described in section 2.3.2) calculated for a single cross-shore section, located at

Lido di Classe, is reported in Fig. 10. The predicted coastal hazard (Fig. 9a) shows that the most critical scenario is in good agreement with the documented coastal impacts, displayed in the right panel. For this event, by comparing hazard forecasts and observations, the use of IWS provides a good prediction (MAX scenario) of coastal impacts for the major part of the Emilia-Romagna coastal area. Moreover, considering the distance between the MIN and the MAX conditions, IWS provides useful information about the range of the possible impacts.

On the 29 October 2018, the City of Venice was inundated by the exceptional sea storm. At 13:40 UTC the sea level reached the peak value of 156 cm at Punta della Salute (fourth historical level of flooding in Venice since 1872), which put three-quarters of the pedestrian public area of the historic town under water. Sea level reaching 120 cm (above local datum), at which point flooding covers 28% of Venice, lasted for about 14 hours. The SHYFEM application to the Venice Lagoon, forced by the open sea TMES results, forecasted sea level peak values of 142 and 161 cm for the MEAN and MAX scenarios, respectively.

Fig. 11 shows the corresponding flooding map of the City of Venice according to the predicted peak values (rounded at 140





and 160 cm). The 59% and 77% of the pedestrian surface are flooded for the two scenarios, respectively. In the first case, the public navigation transport is strongly limited and only links to the islands are guaranteed; besides, most of the elevated walkways are impracticable. In the second case, the public navigation transport is no more guaranteed, as well as all of the elevated walkways. Moreover, many internal channels are no longer navigable due to insufficient vertical space under some

bridges and hence the emergency rescue response may be strongly delayed. Since the observed peak was 156 cm, the MAX scenario provided a realistic prediction of the flooding hazard for the city of Venice.

## 4   Summary and concluding discussion

To improve knowledge on sea storms events in order to progress the prevention and protection measures integrated into coastal defence plan and procedures, we developed a transnational integrated web system (IWS) for sharing observations

and forecasts across the Adriatic and Ionian seas. IWS implementation follows a full-fledged Free and Open Source Software (FOSS) approach, in order to foster transparency, transferability and durability of the system and to be in accord with open source software strategy of the European Commission (European Commission, 2016). IWS provides spatial data infrastructure functionalities for accessing geospatial layers and forecast model outputs through OGC (Open Geospatial Consortium, http://www.opengeospatial.org/) interoperable services. Such approach is widely accepted and implemented at European

(INSPIRE directive, European Commission, 2007; EuroGEOSS initiative, Vaccari et al., 2012) and global level (GEOSS, Global Earth Observation System of Systems) to facilitate intergovernmental and interagency data exchange and harmonization (Maguire and Longley, 2005). Incorporating THREDDS data server, IWS provides access to stored resources also through OPeNDAP and NetCDF standard services and formats. These standards are all products of the scientific communities in oceanography, meteorology and climate sciences and are designed to specifically meet their needs (Hankin et al., 2010), pro-

viding coherent access to a large collection of real-time and archived datasets from a variety of environmental data sources at a number of distributed server site (Unidata, 2019).

It must be taken into account that meteorological and ocean models provide just an approximation of reality, despite their continuous development and improvements. Moreover, the interactions between atmospheric, oceanic and coastal processes are not fully understood, resulting in large uncertainties in the predictions of coastal flooding, in particular, under extreme

conditions (Baart et al., 2011; Zou et al., 2013). This is mainly due to the chaotic nature of the atmosphere and the complexity of the air-sea interactions across scales over several orders of magnitude (Schevenhoven and Selten, 2017). Small errors in the initial conditions of a numerical weather prediction model grow rapidly and affect predictability; forecasted atmospheric conditions are then affected by errors (Molteni et al., 2001). However, as stated by Flowerdew et al. (2010), atmospheric forcing is not the only source of uncertainty in storm surge forecasting. Many other sources of uncertainty, as the model

numerics, resolution, parametrization, boundary conditions and initial sea state, contribute non-linearly to the final forecast uncertainty. The awareness of these uncertainties and prediction errors has led many operational and research flood forecasting systems around the world to move toward numerical forecasts based on a probabilistic concept: the ensemble technique (Cloke and Pappenberger, 2009).



On that basis, the IWS allows to strengthen the forecasts with useful information of their degree of uncertainty and hence analyse the propagation of uncertainty towards the coastal forecasts, starting from the meteorological models. In order to improve sea storm predictions, we implemented for the Adriatic-Ionian macro-region a transnational multi-model ensemble system which combines several existing oceanographic and wave forecasting systems. The magnitude of ensemble spread is

a good indicator of how the forecast accuracy varies between different forecasting situations, indicating a decrease of reliability when the spread increases (World Meteorological Organization, 2012). It is not straightforward what averaging weights should be used for the multi-model ensemble forecast and therefore we used equally weighted members, despite the forecasts which are more precise than others should have more importance in the TMES (Salighehdar et al., 2017; Schevenhoven and Selten, 2017). Here we applied a simple average of the forecasts at every timestamp to compute the ensemble mean, but more

sophisticated methods based on weighting function determined by comparison of the single model results with near real-time observations can be implemented in future (Di Liberto et al., 2011; Salighehdar et al., 2017). Taking advantage of the near real-time observations acquired by the aggregated monitoring network, the root mean square error of the individual forecast will be evaluated and stored for long-term statistics.

Nearshore TMES sea levels and wave characteristics can be directly used in an early warning procedure on the basis of

predefined thresholds for morphological change and for coastal erosion/flooding (e.g., Armaroli et al., 2012). TMES predictions are also used to compute the alongshore total water level time-series. TWL can be used to quantify the vulnerability of the coast to extreme inundation and erosion events, but the estimated run-up values need to be considered with care due to the uncertainty associated to the geomorphological characteristics of the coastal segment units (beach slope in particular). Indeed, Bosom and Jimćnez (2011) and De Leo et al. (2019) found large variability in hazards intensity and vulnerability

along limited coast sectors (20 to 50 km long), even with homogeneous offshore wave conditions. Therefore, the choice of the coastal segment database and its resolution has a substantial effect on the accuracy of the hazard model. The MCD dataset has some limits in reproducing detailed coastal morphologies (i.e. northern Adriatic lagoons and Croatian islands) as well as storm defence structures as breakwaters and seawalls. However, the developed IWS has been designed to be flexible in integrating better defined coastal segment units. If detailed beach geomorphological information is available, the approach of Bosom and

Jimćnez (2011) could be used for assessing the potential of a coastal system to be harmed by the impact of a storm (inundation or erosion), comparing the magnitude of the impact (wave run-up for inundation and beach/shoreline retreat for erosion) with the adaptation capacity of the system (dune/berm height for inundation and beach width for erosion).

The developed system has been tested against observations acquired during a very severe storm that affected the Adriatic Sea on 29 October 2018. TMES forecasts resulted to be in agreement - even if slightly underestimated during the storm

peak - with the observed sea level height and significant wave height. The predicted ensemble mean and standard deviation were combined for creating three different sea condition scenarios all along the Adriatic and Ionian coastline, allowing to evaluate a range of coastal hazard forecast. Moreover, nearshore forecasts were successfully integrated into existing early warning systems for estimating storm hazard at three locations (Slovenia, Emilia-Romagna region, City of Venice). Through this system coupling, the predicted sea conditions were translated into local storm impact indicators and very detailed flooding

maps. The underestimation of predicted sea levels and waves during the 29 October storm peak is probably a consequence



of a general underestimation of the wind forecasts produced by the operational meteorological models. Cavaleri and Bertotti (2004) provided clear evidence of the wind speed problem over the Adriatic Sea. In particular, for the sea storm of the 29 October 2018, Cavaleri et al. (2019) compared the ECMWF model wind with scatterometer data and estimated a 1.11 average enhancement factor.

For the reasons stated above and considering the results presented in this study, the most severe sea condition scenario (MAX = ensemble mean + ensemble standard deviation) can be considered for the investigated area as a realistic and conservative estimation of the peak storm conditions to be used for coastal risk management. Another possible application of TMES outputs could be to use all possible combinations of ensemble mean and standard deviation for the sea level and wave characteristics, providing a large number of sea state combinations. In that way, it would be possible to calculate and estimate the frequency of
exceeding predefined thresholds for coastal hazard. This approach is closer to the idea of the probability of threshold exceeding and will be explored in future.

    The aggregating approach for collecting and sharing observations is crucial for providing real-time information about the sea state - and its evolution - to be used by several countries for prompt emergency response and to increase the overall preparedness to sea storms. Moreover, by merging data from several stations and sensors, IWS is an important storage server
for any data assimilation system. According to Bajo et al. (2019), the assimilation of tide-gauge data in the Adriatic Sea has a strong positive impact on the forecast performance, lasting several days, despite the underestimation in the atmospheric forcing. The forecast improvement is particularly relevant in the case of consecutive sea storms when storm surge levels are influenced by pre-existing oscillations of the basin (seiches) induced by previous events. It is worth mentioning that in the case of the Adriatic Sea - but there could be many other similar situations - the transnational cooperation is crucial for sharing
observations acquired along the whole basin in order to provide real-time information on the sea state to be used in a data assimilation system.

    Real-time observations and numerical model forecasts required to address environmental management problems such as sea storms are commonly excessively intricate for civil protection and stakeholders to use (Magaña et al., 2018). IWS is equipped with geoportal functionalities and interactive geo-visualization tools for simplifying search and access to geospa-
tial data (including forecast model outputs) and monitoring networks time series. Such tools help and assist people who want to use IWS concepts, databases and results in their work and to support their activities. Moreover, a dedicated web site (http://www.seastorms.eu/), designed to foster the data dissemination according to the community-based paradigm and to the Open Data principles (https://opendatacharter.net/), will allow the public data, the forecast results and related statistics to be explored by non-experts over Internet through the use of shared maps, dashboards, graphics, tables and other interactive
geo-visualization tools.

    Concluding, to improve the capacity to react to sea storms, all relevant actors of the coastal area (public authorities, regional and national authorities in charge of Civil Protection, meteorological and forecast services, universities and research institutes, cruise ship enterprises, maritime business enterprises, marinas, aquaculture SMEs, stakeholders from touristic sector) should be involved - through the web and socials - in a transnational cooperation strategy to foster:



- knowledge and data exchange for providing real-time information about the basin-wide sea state through the use of standardized formats, protocols and services;

- coordination of all available ocean forecasting systems in a multi-model ensemble for enhancing the predictability of extreme events and for evaluating the uncertainty of the operational ocean products;

- integration of observations and numerical models through data assimilation techniques for improving the forecast accuracy;

- downscaling of open sea ensemble forecasts to be integrated in site specific early warning systems managed by local authorities;

- data and forecasts dissemination to all relevant coastal actors and citizens over Internet.

*Code and data availability.* The IWS model is available as an open-source code from https://github.com/CNR-ISMAR/iws. The SHYFEM hydrodynamic model is open source and freely available at https://github.com/SHYFEM-model. The data and forecast results used in this study can be accessed, depending on the specific provider's data policy, upon request to the corresponding author. TMES operational forecasts are daily available at http://www.seastorms.eu/.

*Author contributions.* CF conceived the idea of the study with the support of AV, SM and MV. SM and AF designed the IWS structures. SM,
AF, AV, LB, CF, MB, MV, ML and GM prepared the model results and developed the multi-model ensemble. AF, CF and SM developed the scripts for computing TWL on the MCD coastal segments. JJ described the meteorological situation of the 29 October 2018 storm. SU and AV elaborated the TMES results for computing the ER-EWS storm impact indicators for the Emilia-Romagna coast. GM, ES, AP, CF and MB applied the TMES results to the early warning system of the City of Venice. All authors discussed, reviewed and edited the different versions of the manuscript.

*Competing interests.* The authors declare that they have no conflict of interest.

*Acknowledgements.* This work was supported by the I-STORMS project (Integrated Sea sTORm Management Strategies) funded by the European Union under the Interreg V-B Adriatic-Ionian programme with agreement n. 69 of 09-03-2018. The authors wish to thank: Isabella Marangoni, Denise Florean, Anna Zarotti, Alessia Porcu and Silvia Comiati from City of Venice - European Policies Division for promoting and supporting the project activities; Dr. Fabio Raicich from CNR-ISMAR for providing sea level data for the Trieste tide gauge; Dr. Pierluigi
Penna from CNR-IRBIM for providing data for the Senigallia monitoring station; Dr. Luigi Cavaleri and Dr. Luciana Bertotti from CNR-ISMAR for sharing wave results of the Nettuno and Henetus systems; wave data of the Nettuno forecasting systems were kindly provided by Centro Nazionale di Meteorologia e Climatologia Aeronautica (CNMCA); Ing. Maurizio Ferla, Dr. Elisa Coraci and Dr. Roberto Inghilesi



from ISPRA for providing sea level and wave forecasts of the SIMM systems; Janez Polajnar from Slovenian Environment Agency for the data on storm impacts to the Slovenian coast; Dr. Luisa Perini, Dr. Lorenzo Calabrese and Dr. Paolo Luciani from the Geological Seismic and Soil Survey of the Emilia-Romagna Region for providing the report of impacts occurred along the regional coast during the event; Dr. Alessandro Coluccelli and Dr. Aniello Russo from Università Politecnica delle Marche for the support on the AdriaROMS model; the Croatian Hydrographic Institute in Split for the tide-gauge records originating from Rovinj and Dubrovnik and for wave riders records originating from buoys near Rovinj and Dubrovnik/Sv-Andrija.



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

**Figure 1.** Bathymetry of the Adriatic Sea with monitoring stations for sea surface height (yellow dots) and waves (red stars). The 50-year extreme sea levels (ESL) from Vousdoukas et al. (2017) are also reported. Background: EMODnet bathymetry (EMODnet Bathymetry Consortium, 2018).



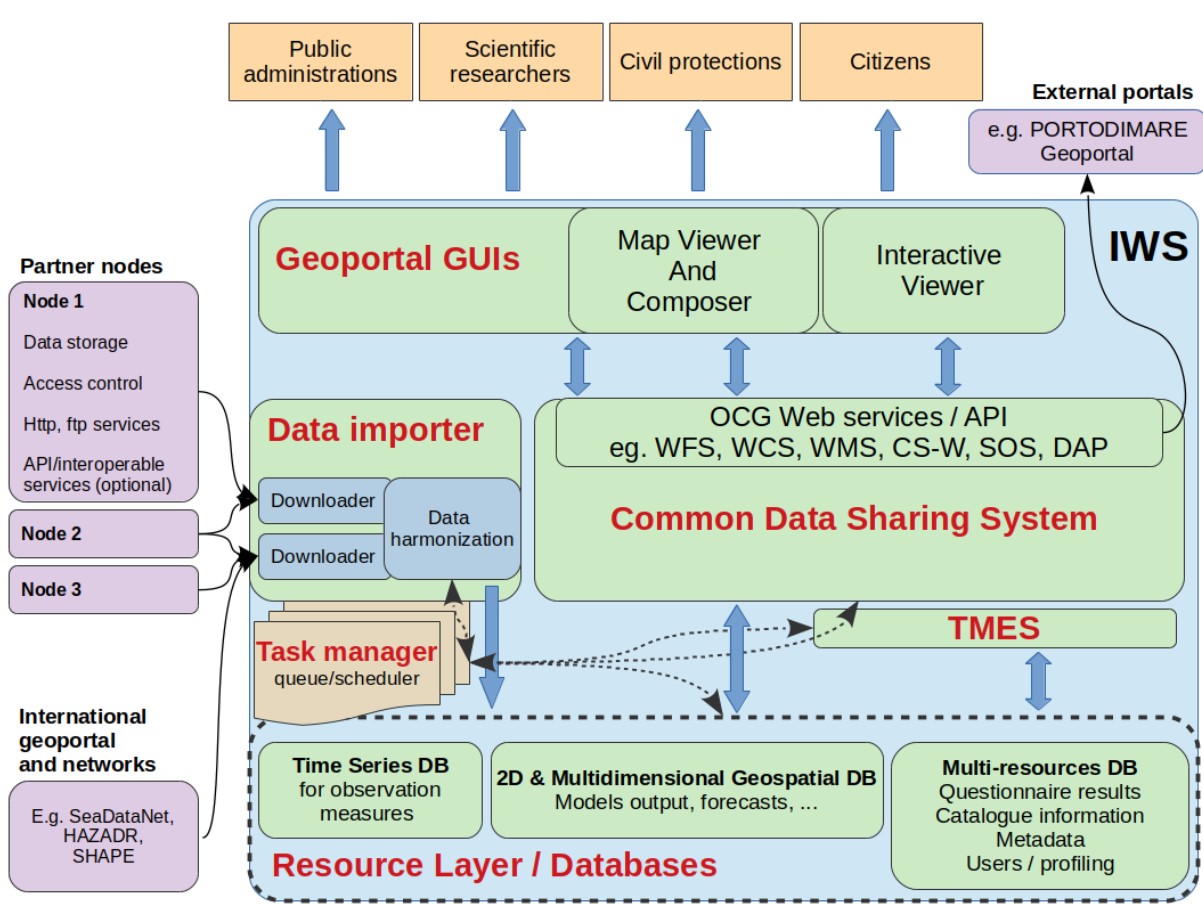

**Figure 2.** Schematic representation of the IWS architecture.

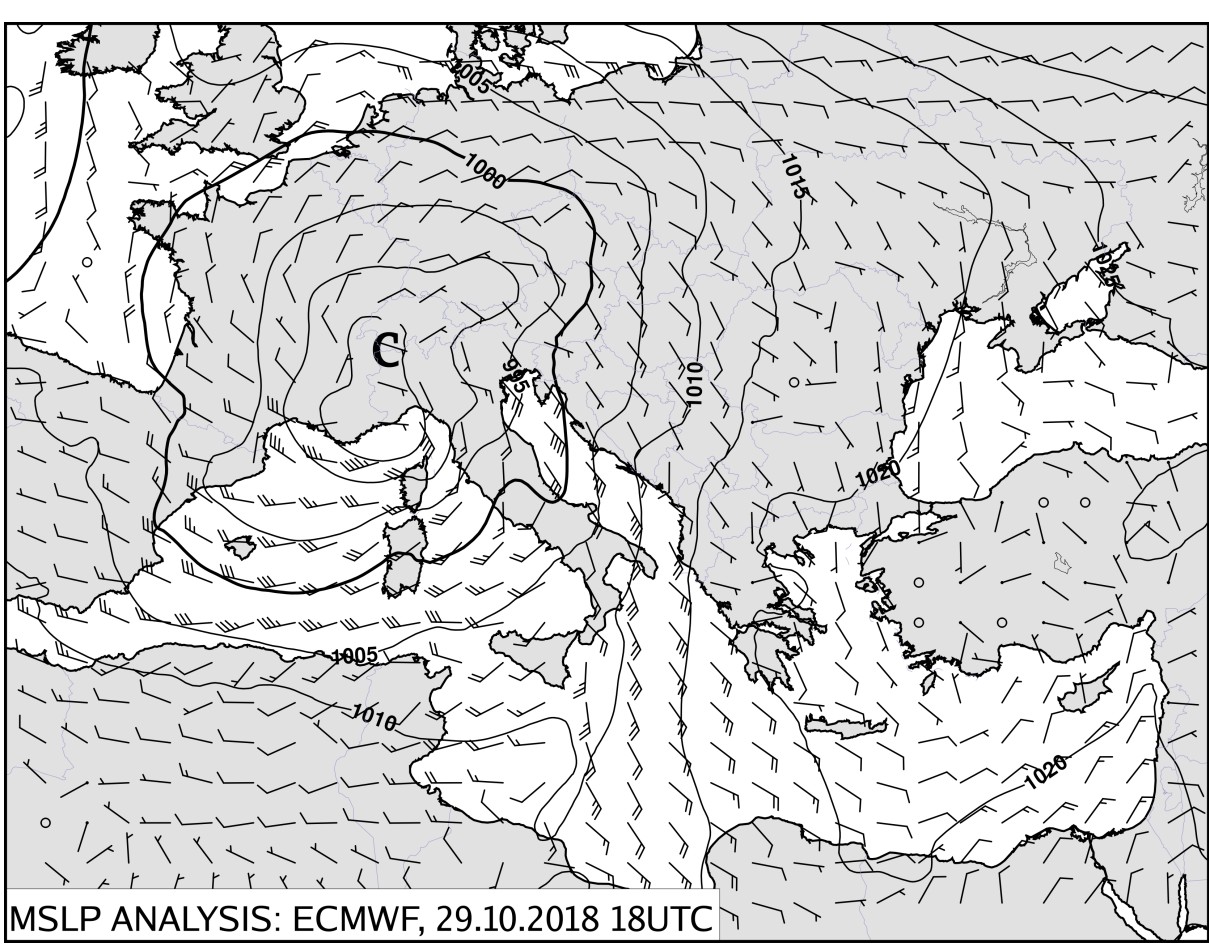

MSLP ANALYSIS: ECMWF, 29.10.2018 18UTC

**Figure 3.** ECMWF 10m wind speed and mean sea level pressure fields over the Mediterranean Sea of 29 October 2018 at 18 UTC.




**Figure 4.** Observed sea level height (a) and significant wave heights (b) measured at different locations (see Fig. 1 for reference).


**Figure 5.** October 2018 results of TMES in terms of the ensemble mean (a, c) and ensemble standard deviation (b, d) for sea level height at 13 UTC (a, b) and significant wave height at 19 UTC (c, d), respectively.



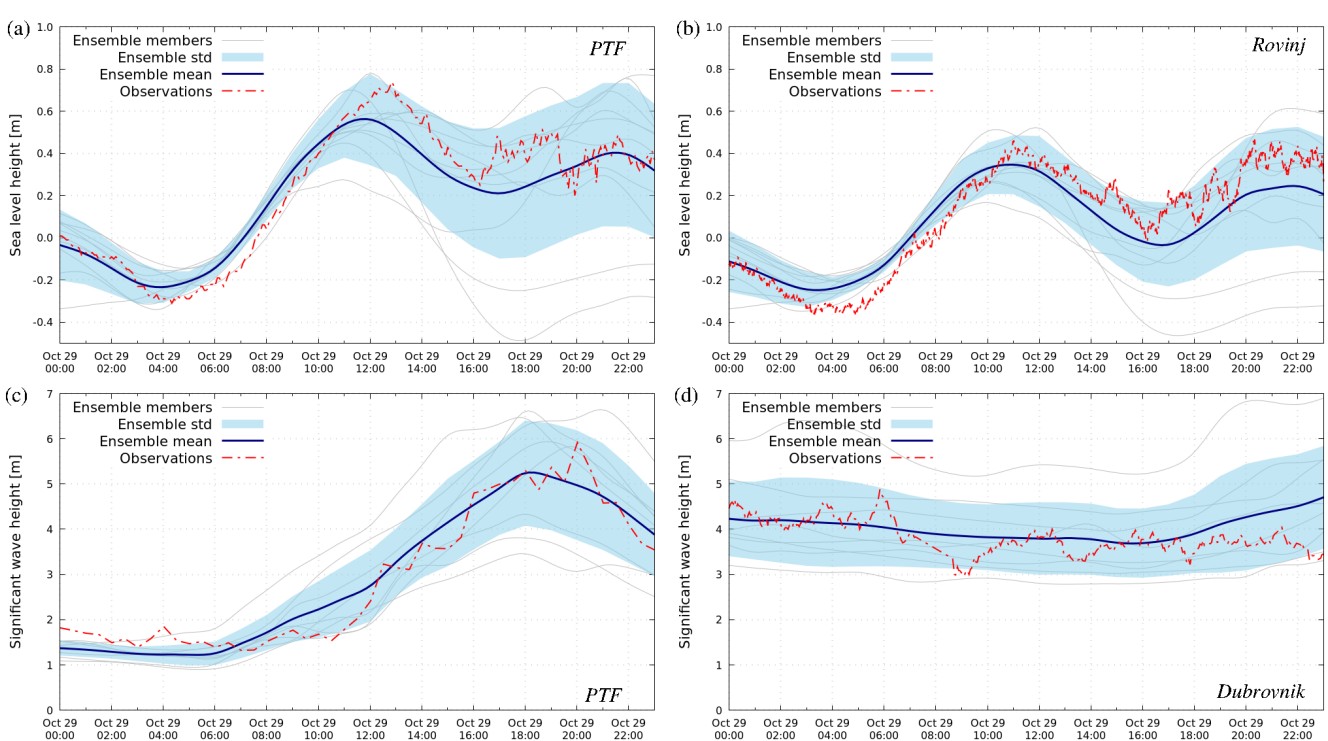

**Figure 6.** TMES sea level height extracted at PTF (a) and Rovinj (b), and significant wave height extracted at PTF (c) and Dubrovnik (d).

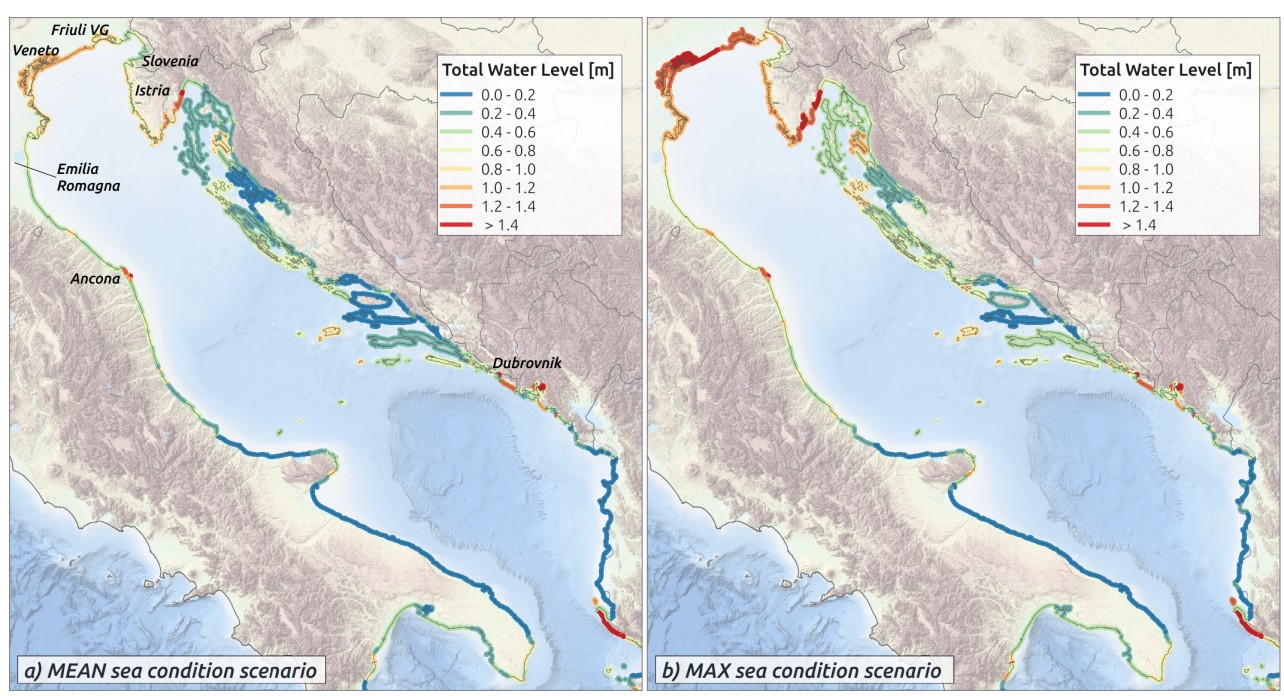

**Figure 7.** Forecasted total water level along the Adriatic coastline for MEAN (a) and MAX (b) sea condition scenarios at 19 UTC of October the 29th, 2018. Background: EMODnet bathymetry (EMODnet Bathymetry Consortium, 2018).

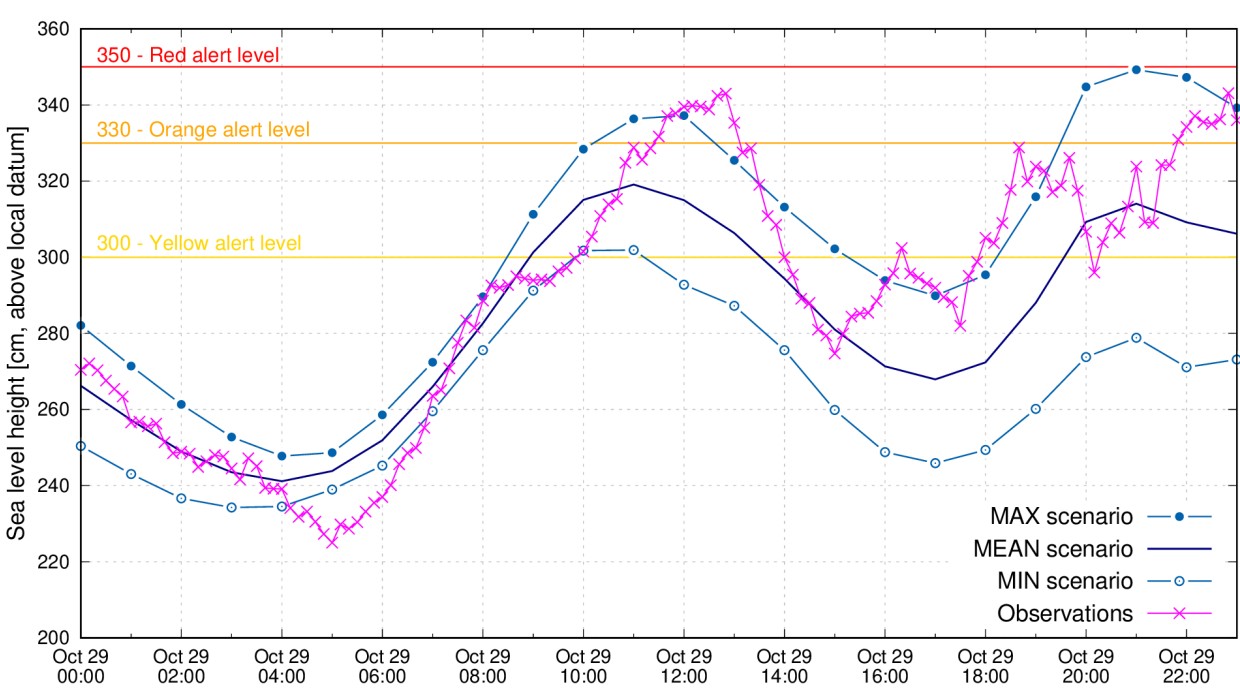

**Figure 8.** Observed and predicted (according to the three sea condition scenarios) sea level height at Koper (Slovenia). The yellow, orange and red lines indicate the adopted thresholds for flooding alerts.



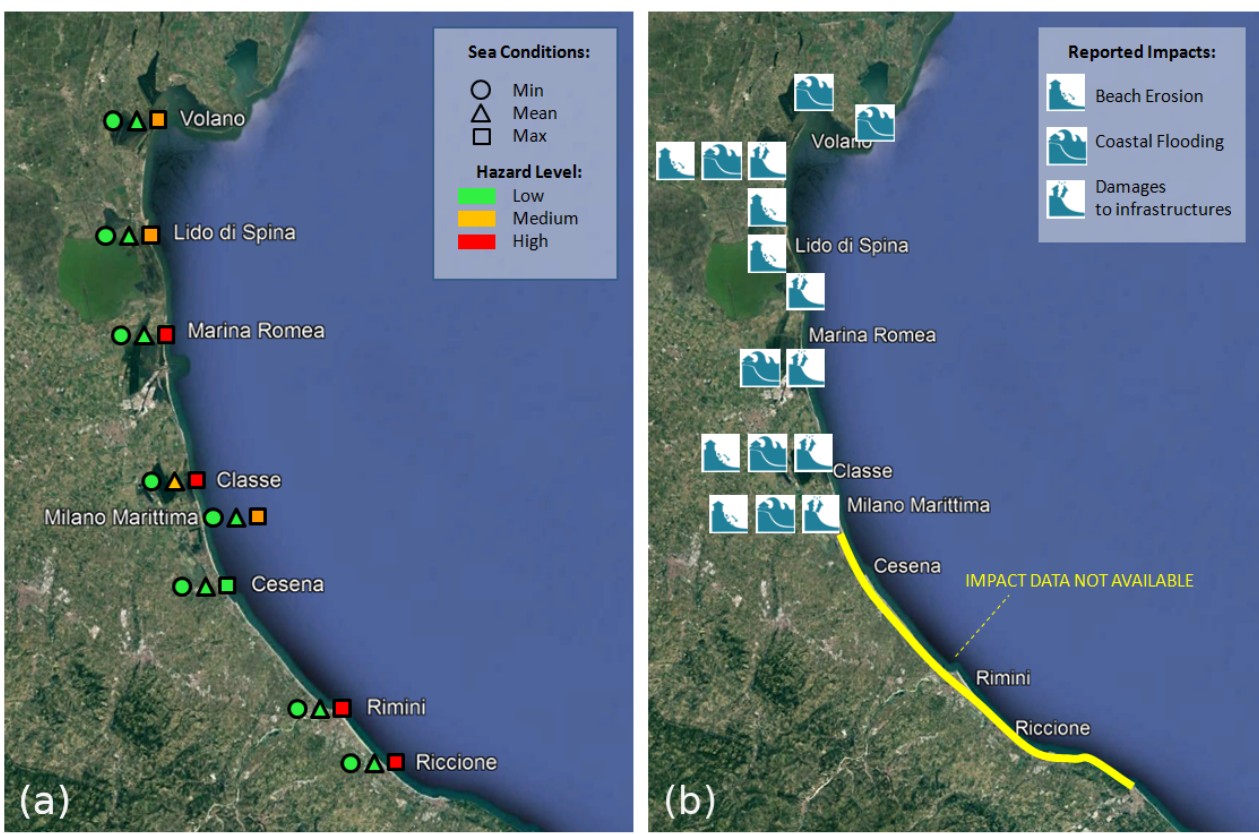

**Figure 9.** Predicted coastal hazard (a) and the documented coastal impacts (b) along the coast of Emilia-Romagna, Italy. Background: image Google, ©2019 TerraMetrics.


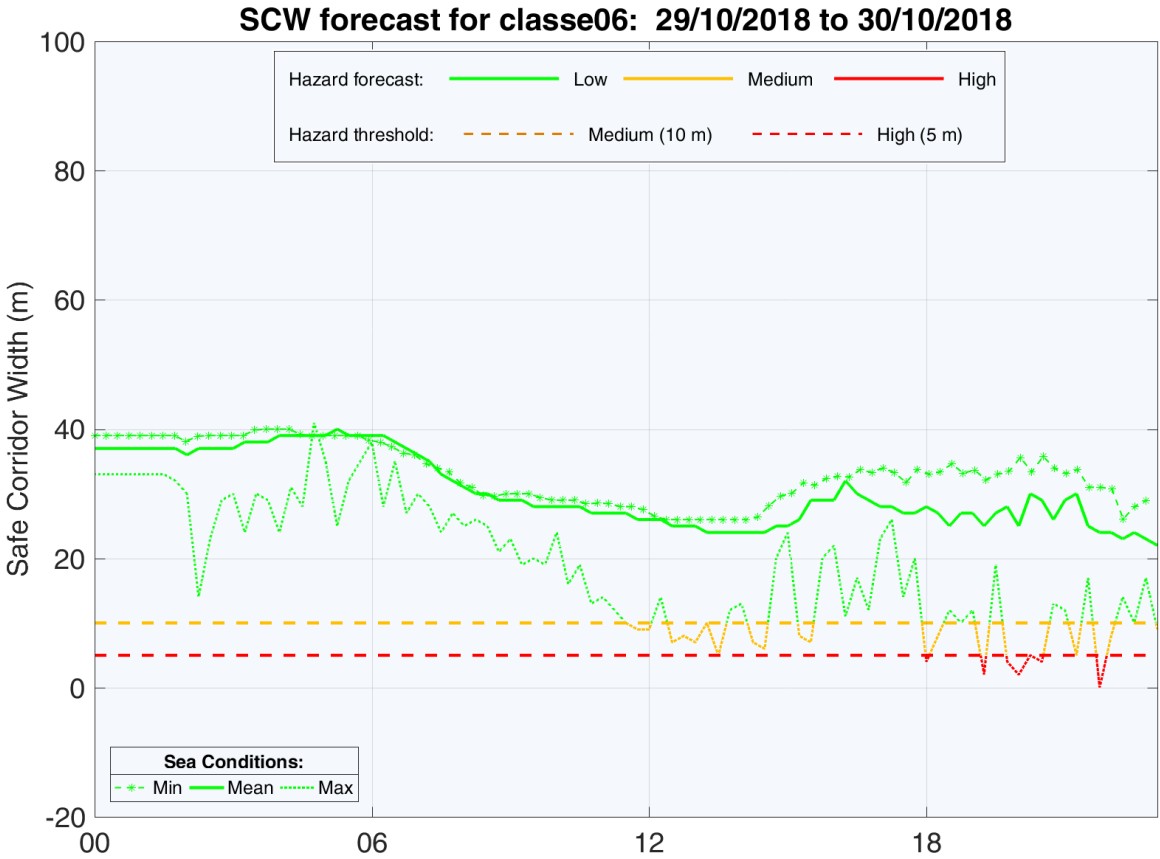

**Figure 10.** Forecasted Safe Corridor Width Index for the beach profile of Classe06 (Lido di Classe, Emilia-Romagna, Italy). The dashed orange and red lines indicate respectively the medium and high thresholds for coastal alerts.

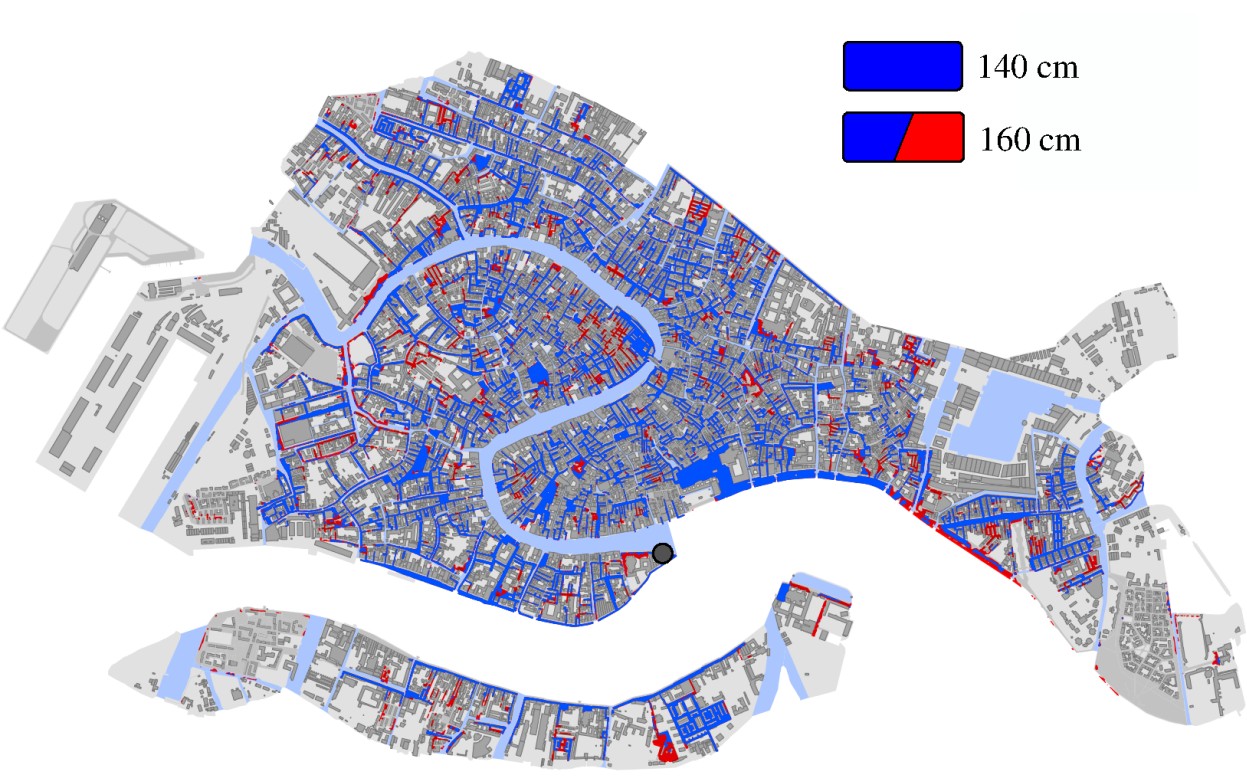

**Figure 11.** Flooding map of the City of Venice according to the predicted sea level height at Punta della Salute (black dot). The colours represent the flooded pedestrian area for sea levels of 140 cm (blue; MEAN scenario) and 160 cm (blue and red; MAX scenarios). Light blue indicates the canals.





**Table 1.** Details of the sea level monitoring stations (yellow dots in Fig. 1).

| Managing authority - Country | Station name | Longitude °E | Latitude °N |
|---|---|---|---|
| City of Venice - IT | Diga Sud Lido | 12.4266 | 45.4182 |
| City of Venice - IT | Diga Nord Malamocco | 12.3414 | 45.3344 |
| City of Venice - IT | Diga Sud Chioggia | 12.3128 | 45.2286 |
| City of Venice - IT | Punta della Salute CG | 12.3364 | 45.4311 |
| City of Venice - IT | Laguna Nord Saline | 12.4719 | 45.4956 |
| City of Venice - IT | Misericordia | 12.3361 | 45.4453 |
| City of Venice - IT | Burano | 12.4175 | 45.4828 |
| City of Venice - IT | Malamocco Porto | 12.2919 | 45.3397 |
| City of Venice - IT | Chioggia Porto | 12.2806 | 45.2325 |
| City of Venice - IT | Chioggia Vigo | 12.2803 | 45.2231 |
| City of Venice - IT | Fusina | 12.2569 | 45.4089 |
| City of Venice - IT | Punta Salute Giudecca | 12.3367 | 45.4306 |
| National Research Council - IT | PTF Acqua Alta | 12.5147 | 45.3231 |
| National Research Council - IT | Meda Abate | 12.7800 | 45.2500 |
| National Research Council - IT | Senigallia | 13.2000 | 43.7500 |
| Ag. for Env. Protection and Energy ER - IT | Porto Garibaldi | 12.2494 | 44.6767 |
| Ag. for Env. Protection and Energy ER - IT | Volano | 12.2742 | 44.7979 |
| Ag. for Env. Protection and Energy ER - IT | Faro | 12.4000 | 44.7900 |
| Inst. for Env. Protection and Research - IT | Trieste | 13.7594 | 45.6469 |
| Inst. for Env. Protection and Research - IT | Ancona | 13.5060 | 43.6246 |
| Inst. for Env. Protection and Research - IT | San Benedetto del T. | 13.8898 | 42.9551 |
| Inst. for Env. Protection and Research - IT | Vieste | 16.1786 | 41.8872 |
| Inst. for Env. Protection and Research - IT | Otranto | 18.4972 | 40.1473 |
| Inst. for Env. Protection and Research - IT | Crotone | 17.1363 | 39.0816 |
| Slovenian Environment Agency - SL | Koper | 13.7245 | 45.5508 |
| Hydrographic Institute - HR | Rovinj | 13.6333 | 45.0833 |
| Hydrographic Institute - HR | Dubrovnik | 18.0677 | 42.6667 |
| Institute of Oceanography and Fisheries - HR | Vela Luka* | 16.7078 | 42.9597 |
| Institute of Oceanography and Fisheries - HR | Starigrad* | 16.5956 | 43.1844 |
| Institute of Oceanography and Fisheries - HR | Sobra* | 17.6006 | 42.7444 |
| Institute of GeoSciences - AL | Vlore Triport | 19.3936 | 40.5144 |
| Institute of GeoSciences - AL | Durres | 19.4526 | 41.3025 |
| Institute of GeoSciences - AL | Vlore | 19.4810 | 40.4501 |
| Institute of GeoSciences - AL | Sarande | 20.0035 | 39.8705 |
| Institute of GeoSciences - AL | Shengjin | 19.5854 | 41.8124 |

*Available through http://www.ioc-sealevelmonitoring.org/.





**Table 2.** Details of the wave monitoring stations (red stars in Fig. 1).

| Managing authority - Country | Station name | Longitude °E | Latitude °N |
|---|---|---|---|
| City of Venice - IT | Misericordia | 12.3361 | 45.4453 |
| National Research Council - IT | Senigallia | 13.2000 | 43.7500 |
| National Research Council - IT | PTF Acqua Alta | 12.5147 | 45.3231 |
| Ag. for Env. Protection and Energy ER - IT | Nausicaa | 12.4766 | 44.2155 |
| National Institute of Biology - SL | Vida | 13.5454 | 45.5508 |
| Slovenian Environment Agency - SL | Zora | 13.6717 | 45.6033 |
| Slovenian Environment Agency - SL | Zarja | 13.5354 | 45.6016 |
| Hydrographic Institute - HR | Rovinj | 13.5156 | 45.0736 |
| Hydrographic Institute - HR | Dubrovnik | 17.9550 | 42.6467 |

**Table 3.** Details of the sea level forecasting systems used in the TMES. Key references are reported at the bottom of the table.

| Managing authority - Country | System name | Domain | Horizontal res. | Core engine | Tide | Baroclinic | Meteo forcing |
|---|---|---|---|---|---|---|---|
| City of Venice - IT | SHYMED[1] | Mediterranean Sea | var. up to 200 m | SHYFEM | no | no | ECMWF |
| National Research Council - IT | Kassandra[2] | Mediterranean Sea | var. up to 100 m | SHYFEM | yes | no | BOLAM, MOLOCH |
| National Research Council - IT | ISSOS | Mediterranean Sea | var. up to 200 m | SHYFEM | no | no | BOLAM |
| National Research Council - IT | Tiresias[3] | Adriatic Sea | var. up to 10 m | SHYFEM | yes | yes | MOLOCH |
| Ag. for Env. Protection and Energy ER - IT | AdriaROMS[4] | Adriatic Sea | 2 km | ROMS | yes | yes | COSMO-5M |
| Ag. for Env. Protection and Energy ER - IT | Adriac[5] | Adriatic Sea | 1 km | ROMS | yes | yes | COSMO-2I,5M |
| Inst. for Env. Protection and Research - IT | SIMMb[6] | Mediterranean Sea | var. up to 1 km | SHYFEM | no | no | BOLAM |
| Inst. for Env. Protection and Research - IT | SIMMe[6] | Mediterranean Sea | var. up to 1 km | SHYFEM | no | no | ECMWF |
| Slovenian Environment Agency - SL | SMMO[7] | Adriatic Sea | 1/72; 1/216 deg | NEMO | yes | yes | ALADIN |
| CMCC - IT | MFS[8] | Mediterranean Sea | 1/24 deg | NEMO | yes | yes | ECMWF |

[1]Bajo and Umgiesser (2010); [2]Ferrarin et al. (2013); [3]Ferrarin et al. (2019); [4]Russo et al. (2013); [5]Bressan et al. (2017); [6]Mariani et al. (2015); [7]Ličer et al. (2016); [8]Tonani et al. (2009).

**Table 4.** Details of the wave forecasting systems used in the TMES. Key references are reported at the bottom of the table.

| Managing authority - Country | System name | Domain | Horizontal res. | Core engine | Meteo forcing |
|---|---|---|---|---|---|
| National Research Council - IT | Kassandra[1] | Mediterranean Sea | var. up to 100 m | WWMIII | BOLAM, BOLAM |
| CNMCA / National Research Council - IT | Nettuno[2] | Mediterranean Sea | 4.5 km | WAM | COSMO-ME |
| National Research Council - IT | Henetus[3] | Adriatic Sea | 1/12 deg | WAM | ECMWF |
| Ag. for Env. Protection and Energy ER - IT | SWAN[4] | Med., Adriatic Sea | 25 km, 8 km | SWAN | COSMO-5M |
| Ag. for Env. Protection and Energy ER - IT | Adriac | Adriatic Sea | 1 km | SWAN | COSMO-2I,5M |
| Inst. for Env. Protection and Research - IT | SIMM[5] | Med., Adriatic Sea | 1/30; 1/240 deg | WAM, SWAN | ECMWF, BOLAM |
| Slovenian Environment Agency - SL | SMMO[6] | Central Med. Sea | 1/60 deg | WAM | ALADIN |
| Met. and Hydrol. Serv. - HR | WWM[7] | Adriatic Sea | var. up to 10 m | WWM | ALADIN |
| HCMR - GR | MED-Waves[8] | Mediterranean Sea | 1/24 deg | WAM | ECMWF |

[1]Ferrarin et al. (2013); [2]Bertotti et al. (2013); [3]Bertotti et al. (2011); [4]Valentini et al. (2007); [5]Mariani et al. (2015); [6]Ličer et al. (2016); [7]Dutour Sikirić et al. (2018); [8]Zacharioudaki et al. (2015).