# Peer review of "Integrated sea storm management strategy: the 29 October 2018 event in the Adriatic Sea"

_Natural Hazards and Earth System Sciences, 2019_

## Referee Comment (RC1) · Anonymous Referee #1 · 21 Aug 2019

This manus describes a collaborative effort between agencies in Italy, Croatia and Slovenia to develop an integrated system for storm handling. A Transnational Multi-model Ensemble System (TMES) is used to handle the input from a number of operational model systems. The severe storm of 29 October 2018 is used as a convincing example of the usefulness of the new system. I have one comment: Why are no products from the Copernicus Marine Environment Monitoring System applied and part of the TMES? They are readily available and could provide an even more robust ensemble system.

---

## Author Comment (AC1) · 27 Aug 2019

We thank the reviewer for the positive comment. The developed Transnational Multimodel Ensemble System (TMES) integrates also some Copernicus Marine Environment Monitoring System (CMEMS) products. In details, the Mediterranean Forecasting System (managed by CMCC) and the MED-Waves (managed by HCMR) results are used in sea level height and the wave multi-model ensembles, respectively (see Tables 3 and 4 in the manuscript). In the revised manuscript, we will enhance the description of the operational forecasting systems used in the TMES and clarify that the mentioned products are retrieved from CMEMS.

2019-212, 2019.

---

## Referee Comment (RC2) · Anonymous Referee #2 · 27 Sep 2019

General comments: I have just finished reviewing the manuscript entitles 'Integrated sea storm management strategy: the 29 October 2018 event in the Adriatic Sea' by Ferrarin et al. Overall the article provides a good description of system that combines information from different forecasting models in order to provide an ensemble prediction. I believe that the article is of interest for the journal. It has a high technical character that is justifiable by the nature of the topic. However, I believe that in order for the manuscript to be accepted some significant changes need to be addressed. There are some fundamental misuse of the risk terminology that need further clearance this is principally in the introduction bust also thought the document (see detailed comments). Introduction needs improvement especially the first part with the incorporation of objectives of the study. Some interesting information of the state of the art is

placed in conclusion but I think is more fitted for the introduction. A more detail comparison of the individual models and the model ensemble with observed data would be very interesting.

Specific comments:

2.2 Forecasting System: Given technical character and it is describing a complex method of forecast ensemble I believe that Tables 2 and 3 can be expanded to include details indicating the forecast characteristics (e.g. forecast window, update time) would be very interesting for some readers. Also the meteorological models resolution are interesting, some data are already given in the text but I believe that a thorough description could be useful. I also believe that Table 1 and 2 are not very interesting and good be eliminated or moved to an appendix.

2.3 Forecasting System: It is not clear if the IWS system is providing or receiving detail EWS. It would be useful to present the EWS in Figure 2. I believe that Figure 2 needs to be improved to provide a more detail presentation of the system. What about the other areas of the Adriatic that are not covered by the EWSs. Is a hazard map produced and how what kind of topographic information is used?

3.2 Storm Predictability: The section is well structured and with some interesting figures, however a more detailed analysis and statistical representation of the individual model and the model ensemble should be presented. An interesting question especial for regional assembles is the spatial performance of the different model. Although Fig 5 present a good synoptic view of the ensemble performance a more detailed look (Fig6) reviles that there are two models that substantially under predict the sea level height for the final part of the storm. Why is this happening, are there any performance criteria for the models to enter the ensemble?

3.3 Storm Hazard and Impact assessment on the coast: Is the model ensemble always under-estimates the events? Do you have other examples that also indicate the MEAN+STD is a better estimate for the events? Related with the hazard the authors

mention that the calculated on each coastal assessment unit. Some more information on how many units were identified in the area and how they are distributed it would be interesting for the reader. It is unclear if the Stockton model is used in all areas. If yes this is contradicting the previous comments of the authors. Finally, the detail description of the storm in the study area gives valuable information however a directed comparison of the hazard intensity and extend predicted by the ensemble and the one observed is missing. Such comparison is important in order to identify the advantages of such a model.

4 Summary and Concluding discussion: The start of this section gives valuable detail information of the system that they should be placed in section 2.3.

Technical corrections:

Page 2 - Line1: 'Sea storm . . . directly impact on the citizens quality of life '. This statement is not exactly true in my opinion. Sea storms are a natural phenomenon and they do not affect the quality of life. Possible risks associated with sea storms can have this effect. The second paragraph provides a description of the process based models and it is more suited for the introduction. Page 2 - Line14: The reference of Roland et al., 2009 is not appropriate for wave setup maybe an older reference would be more appropriate (e.g. Longuet-Higgins, M.S., Stewart, R.W., 1963. A note on wave set-up. Journal of Marine Research 21, 4-10.) Page 2 - Line14: 'they travel up and down the beach'. Are you referring to swash processes? Page 2 - Line16-19: 'Coastal flooding, erosion, impacts on ecosystems, damages to infrastructures and productive activities can worsen if combined with the absence of adequate early warning systems, coordinated strategies, intervention procedures, coastal management and planning with significant related economic costs' . This sentence is mixing hazards and consequences with primary measures and management strategies. This can results in a confusion of terms that is undisariable. For example coastal flooding and erosion are not related with EWS. The absence of an EWS can results in increased damages if proper disaster risk reduction measures (DRRs) are not implemented. I suggest to restructure the

sentence.

Page 3 and throughout the document: The terms ′Bora′ and ′Sirocco′ are local wind names it is better to use italic fond style or directly use the English name. Figure 2 : What is the difference between the black thin lines the blue arrow and the dash line? Why the TMES exchanges information only with the Recourse layer (maybe a better explanation of what the TMES is doing could we useful). Only the resources layer is delimited I think it would be nice to show all 6 layers limits. Page 7 – Line6-7: 'hazard maps . . . to identify vulnerable areas' there is a mix of the two terms that is common but not a good practice. I suggest following UNISDR Terminology where hazard is related with the physical aspects of the storm the coastal area and vulnerability with the socioeconomic aspects.

Page 7 – Line25: 'It must be pointed out that the widely used Stockdon's. . .' The Stockdon formula is not applicable in rocky and gravel beaches. The problem is not the underestimation of the runup is the use of an inappropriate formula that results in underestimation of the runup. Figure 3: Substitute the 'C' by 'L' for low pressure Page 12 – Line11-13: There is a large number of local names that are not shown in the figure and is difficult to follow by the reader. Please add a more detail figure. Page 14 – Line27: 'adaptation capacity' the dune and berm characteristics of a beach are not the adaptation capacity. The term adaptation is related with the ability of the system to overcome long term changes in forcing factors. Beach and berm characteristics can be combined with physical parameters (e.g. wave height water level) to calculate process based indexes that can serve as hazard intensity and extend parameters. A review of such indexes can be found in Ferreira, Ó., Plomaritis, T.A., Costas, S., 2017. Process-based indicators to assess storm induced coastal hazards. Earth-Science Reviews 173, 159-167.

---

## Author Response (AR1)

**Responses to the Editor's and Reviewers' Comments and Suggestions**

Journal: Natural Hazards and Earth System Sciences (NHESS)

Manuscript number: nhess-2019-212

Manuscript title: Integrated sea storm management strategy: the 29 October 2018 event in the Adriatic Sea

We would like to thank the Editor and Reviewers for their valuable comments and effort to improve the manuscript. We have responded to all comments as can be seen in the following list. We believe that with these revisions the manuscript has been improved and we hope that it is now ready for publication.

The original Editor's and Reviewers' comments and suggestions are shown in regular typeface, while our responses are shown in italics. The line and figures numbers we use refer to the revised document.

**Response to the Editor**

**E1** a) I agree that you may not include in this study a detailed evaluation of the individual model performances (see reviewer 2), but I suggest that you insert a paragraph describing how to deal with the different accuracies of the models in a future operational framework.

*Response: Following the Editor's suggestion, we included a paragraph in the "Summary and concluding discussion" (Page 14, lines 15-22) describing the future development to account for the different accuracies of the models in the multi-model ensemble. The sentence reads: "It is not straightforward what averaging weights should be used for the multi-model ensemble forecast and therefore we used equally weighted members, despite the forecasts which are more precise than others should have more importance in the TMES (Salighehdar et al., 2017; Schevenhoven and Selten, 2017). Here we applied a simple average of the forecasts at every timestamp to compute the ensemble mean, but more sophisticated methods based on weighting function determined by comparison of the single model results with near real-time observations can be implemented in future (Di Liberto et al., 2011; Salighehdar et al., 2017). Taking advantage of the near real-time observations acquired by the aggregated monitoring network, the root mean square error of the individual forecast will be evaluated and stored for long-term statistics." See also the response to comment R2.4.*

**E2** Further, I note that there are two issues that are relevant for early warning systems and are likely important component in an operational framework. They are briefly mentioned in the "Summary and Conclusions", but you do not mention the former studies already available in the literature, specifically for Venice:

   b) the inclusion of a data assimilation procedure (e.g. Lionello et al., 2006)

   c) the probabilistic approach provided by a single model ensemble prediction. Note that papers showing this specifically for Venice are already available (e.g. Mel and Lionello, 2014a,b, 2016).

I suggest you paper your text is updated including these (and other) references.

*Response: We thank the editor for bringing these useful citations to our attention. We cited the study of Lionello et al. (2006) at Page 15 (line 25) when discussing the importance of data assimilation for storm surge forecasting in the Adriatic Sea. At Page 14 (lines 6-13), we extended the discussion of the probabilistic forecast by including the following sentence regarding single model ensemble prediction: "The awareness of the prediction uncertainties and errors has led many operational and research flood forecasting systems around the world to move toward numerical forecasts based on a probabilistic concept: the ensemble technique (Cloke and Pappenberger, 2009). In this contest, a probabilistic forecasting system could be based on perturbation of initial conditions, forcing and parameters of a single model (Flowerdew et al., 2010; Bernier and Thompson, 2015; Salighehdar et al., 2017). Such approach has been already applied to the Adriatic Sea for improving storm surge forecast and providing a realistic estimate of the prediction uncertainty (Mel and Lionello, 2014a,b, 2016; Bajo et al., 2019)."*

**Response to Reviewer #1**

**R1.1** This manus describes a collaborative effort between agencies in Italy, Croatia and Slovenia to develop an integrated system for storm handling. A Transnational Multimodel Ensemble System (TMES) is used to handle the input from a number of operational model systems. The severe storm of 29 October 2018 is used as a convincing example of the usefulness of the new system. I have one comment: Why are no products from the Copernicus Marine Environment Monitoring System applied and part of the TMES? They are readily available and could provide an even more robust ensemble system.

*Response: We thank the reviewer for the positive comment. The developed Transnational Multimodel Ensemble System (TMES) integrates also some Copernicus Marine Environment Monitoring System (CMEMS) products. In details, the Mediterranean Forecasting System (managed by CMCC) and the MED-Waves (managed by HCMR) results are used in sea level height and the wave multi-model ensembles, respectively (see Tables 1 and 2 in the manuscript). In the revised manuscript (Page 7, line 6), we enhanced the description of the operational forecasting systems used in the TMES and clarified that some products are retrieved from CMEMS.*

**Response to Reviewer#2**

**R2.1** General comments: I have just finished reviewing the manuscript entitles "Integrated sea storm management strategy: the 29 October 2018 event in the Adriatic Sea" by Ferrarin et al. Overall the article provides a good description of system that combines information from different forecasting models in order to provide an ensemble prediction. I believe that the article is of interest for the journal. It has a high technical character that is justifiable by the nature of the topic. However, I believe that in order for the manuscript to be accepted some significant changes need to be addressed. There are some fundamental misuse of the risk terminology that need further clearance this is principally in the introduction but also thought the document (see detailed comments). Introduction needs improvement especially the first part with the incorporation of objectives of the study. Some interesting information of the state of the art is placed in conclusion but

I think is more fitted for the introduction. A more detail comparison of the individual models and the model ensemble with observed data would be very interesting.

*Response: We appreciate the comments and we improved the manuscript following all reviewer's suggestions. We corrected the risk terminology over the whole document (see the responses to comments R2.7, R2.10 and R2.13). Some information we wrote in the "Conclusions" at first, had been repositioned within the "Introduction" and "Material and methods" (see also the response to comment R2.6). The aim of the study is presented at the end of the introduction section.*

*This manuscript aims at presenting the structure for sharing knowledge, data and forecasts in order to improve the prevention and protection measures to sea storm emergencies. Therefore, despite the multi-model ensemble system is a fundamental component of the developed systems, we decided not to include in this study a detailed evaluation of the individual models performance. The comparison of the models and the ensemble with observed data will be the subject of a future work which will consider the analysis of a more complete dataset of sea storm events (see also the response to comment R2.4).*

**R2.2** Forecasting System: Given technical character and it is describing a complex method of forecast ensemble I believe that Tables 2 and 3 can be expanded to include details indicating the forecast characteristics (e.g. forecast window, update time) would be very interesting for some readers. Also the meteorological models resolution are interesting, some data are already given in the text but I believe that a thorough description could be useful. I also believe that Table 1 and 2 are not very interesting and good be eliminated or moved to an appendix.

*Response: Following the reviewer's suggestion we included the forecast range and the meteorological models resolution in Table 3 and 4 (now Table 1 and 2). The tables providing the monitoring stations details have been moved to the appendix.*

**R2.3** Forecasting System: It is not clear if the IWS system is providing or receiving detail EWS. It would be useful to present the EWS in Figure 2. I believe that Figure 2 needs to be improved to provide a more detail presentation of the system. What about the other areas of the Adriatic that are not covered by the EWSs. Is a hazard map produced and how what kind of topographic information is used?

*Response: As described in section 2.3 and 3.3, IWS had been designed to provide multi-model forecast products to existing early warning systems, developed in areas were a deep knowledge of the coastal dynamics and high-resolution datasets (topography and bathymetry) are available. These concept were clarified in the manuscript (Page 5 line 7, Page 8 lines 20-21). Moreover, following the reviewer's suggestion we improved Figure 2.*

*The multi-model results have been used to provide a basin-wide overview of the physical processes acting in coastal areas and responsible for storm related hazards. TMES products are combined with the coastal characteristics (coast material and slope) provided by the MCD database for computing the total water level (TWL). For the coastal segments characterized by sandy beaches, TWL was computed combining the sea level height, wave setup and wave runup according to the Stockdon's formula ($R_2$, the 2% exceedance level of runup maxima; Stockdon et al., 2006). The resulting maps (Fig. 7) provide a basin-wide overview of the physical processes acting in coastal areas and responsible for storm related hazards. The forecasted TWLs are made available to the IWS users and can be combined with a digital elevation model (DEM) of the coast for estimating inundation intensity and extend. See also the responses to comments R2.5 and R2.14.*

**R2.4** Storm Predictability: The section is well structured and with some interesting figures, however a more detailed analysis and statistical representation of the individual model and the model ensemble should be presented. An interesting question especial for regional assembles is the spatial performance of the different model. Although Fig 5 present a good synoptic view of the ensemble performance a more detailed look (Fig6) reviles that there are two models that substantially under predict the sea level height for the final part of the storm. Why is this happening, are there any performance criteria for the models to enter the ensemble?

*Response: The multi-model ensemble forecasting model was created with the aim of combining together the existing available operational systems without providing a critical review of the individual model performance. The storm event of $29^{th}$ October 2018 is here taken as a pilot study for applying and testing the developed approach. At the same time, the analysis of the model results provide an example of the scatter of the individual forecasts (ensemble standard deviation) to point the attention on the uncertainty of the sea condition prediction. To our opinion, the awareness of these uncertainties and prediction errors is crucial and should be accounted for in managing coastal risks related to sea storms.*

*Actually, there is no performance criteria for the models to enter the ensemble, but, as discussed at Page 14 (lines 15-22), weighting function determined by comparison of the single model results with observations will be implemented in future in the TMES. Therefore, the forecasts which are more precise than others will have more importance in the multi-model ensemble (see also the response to comment E1.*

**R2.5** Storm Hazard and Impact assessment on the coast: Is the model ensemble always under-estimates the events? Do you have other examples that also indicate the MEAN+STD is a better estimate for the events? Related with the hazard the authors mention that the calculated on each coastal assessment unit. Some more information on how many units were identified in the area and how they are distributed it would be interesting for the reader. It is unclear if the Stockton model is used in all areas. If yes this is contradicting the previous comments of the authors. Finally, the detail description of the storm in the study area gives valuable information however a directed comparison of the hazard intensity and extend predicted by the ensemble and the one observed is missing. Such comparison is important in order to identify the advantages of such a model.

*Response: We thank the reviewer for the useful comment, which helped us to clarify the approach adopted in this study. In this manuscript, we present the results of the TMES application to only one storm event and therefore we cannot indicate that the MAX (MEAN+STD) sea condition scenario is always the better estimation. It is however true that, generally, the wind condition over the Adriatic Sea are underestimated by many meteorological models (Cavaleri and Bertotti, 2004; Cavaleri et al., 2019). For these reasons, the most severe sea condition scenario can be considered for the investigated area as a conservative estimation of the peak storm conditions to be used for coastal risk management. As stated in the response to comment R2.1, the comparison of the models and the ensemble with observed data related to a more complete dataset of sea storm events will be the subject of a future work.*

*We clarified that, in order to assess the perception of the physical processes acting in coastal areas and responsible for storm related hazards, the coast is subdivided into segments of variable length in function of morphology, human settlements and administrative boundaries. The coastal assessment units were selected according to the Mediterranean Coastal Database (MCD) developed by Wolff et al. (2018). We let the reader to refer to*

*above-cited article for more details about how the coastal units were defined. The MCD segments have an average length of 4.5 km. However, as discussed at Page 14 lines 29-35, the MCD segments are sometimes too coarse to represent complex morphologies, especially in confined coastal systems (lagoons) and along the eastern rocky coast.*

*The Stockdon's formula was improperly applied to all coastal segments. As specified in the response to comment R2.14, we corrected the manuscript specifying that the Stockdon's formula is applied only to the coastal segments characterised by sandy beaches (Page 8, lines 15-17).*

*We do agree with the reviewer that a more detailed comparison between predicted and observed hazard would add a significant contribution to the presented results. Unfortunately, detailed coastal observations of the hazard intensity and extend are not available. For the Emilia-Romagna and Slovenia coasts, flooding and erosion were reported by the local authorities and therefore the comparison between predicted and observed is qualitative and not quantitative. In the case of the City of Venice, the situation is similar, even if a more detailed information of the extend of the flooding is available as function of the sea level (see Figure 11). The flooding maps reported in the manuscript were obtained by the municipality imposing the sea level height observed/predicted at Punta della Salute (at intervals of 10 cm) to a centimetre accurate digital terrain model of the city.*

**R2.6** Summary and Concluding discussion: The start of this section gives valuable detail information of the system that they should be placed in section 2.3.

*Response: Following the reviewer's suggestion we moved part of the first paragraph of the "Summary and Concluding discussion" section to the "Material and methods" section.*

**R2.7** Page 2 - Line 1: "Sea storm ... directly impact on the citizens quality of life". This statement is not exactly true in my opinion. Sea storms are a natural phenomenon and they do not affect the quality of life. Possible risks associated with sea storms can have this effect. The second paragraph provides a description of the process based models and it is more suited for the introduction.

*Response: We reformulated the first paragraph on the introduction as follow (Page 2, lines 2-9): "Sea storms represent the main threat in coastal areas. In fact, they can cause a range of potential hazards, such as coastal erosion and inundation, as well as damages to infrastructure and to the important cultural heritage exposed to these phenomena (Chaumillon et al., 2017; Reimann et al., 2018; Vousdoukas et al., 2018). Along the coast, extreme storms can also significantly affect businesses activities, such as aquaculture, fisheries, tourism and beach facilities. The potential future ...".*

**R2.8** Page 2 - Line 14: The reference of Roland et al., 2009 is not appropriate for wave setup maybe an older reference would be more appropriate (e.g. Longuet-Higgins, M.S., Stewart, R.W., 1963. A note on wave set-up. Journal of Marine Research 21, 4-10.).

*Response: Following the reviewer's suggestion we replaced Roland et al. (2009) with the more appropriate Longuet-Higgins and Steward (1963).*

**R2.9** Page 2 - Line 14: "they travel up and down the beach". Are you referring to swash processes?

*Response: Yes, we refer to swash processes. We modified the sentence (Page 2, line 14) as follow: " ... they travel up and down the beach (swash processes)".*

**R2.10** Page 2 - Line 16-19: "Coastal flooding, erosion, impacts on ecosystems, damages to infrastructures and productive activities can worsen if combined with the absence of adequate*

early warning systems, coordinated strategies, intervention procedures, coastal management and planning with significant related economic costs". This sentence is mixing hazards and consequences with primary measures and management strategies. This can results in a confusion of terms that is undisariable. For example coastal flooding and erosion are not related with EWS. The absence of an EWS can results in increased damages if proper disaster risk reduction measures (DRRs) are not implemented. I suggest to restructure the sentence.

*Response: We concur with the reviewer that the mentioned sentence was not properly formulated. We modified it as follow (Page 2, lines 30-32): "Coastal flooding of urban areas, beach erosion, damages to infrastructures and productive activities can worsen if combined with the absence of an adequate sea storm management strategy with significant related economic costs."*

**R2.11** Page 3 and throughout the document: The terms "Bora" and "Sirocco" are local wind names it is better to use italic font style or directly use the English name.

*Response: Following the reviewer's suggestion we used italic font style for the terms Bora and Sirocco.*

**R2.12** Figure 2: What is the difference between the black thin lines the blue arrow and the dash line? Why the TMES exchanges information only with the Recourse layer (maybe a better explanation of what the TMES is doing could we useful). Only the resources layer is delimited I think it would be nice to show all 6 layers limits.

*Response: According to the reviewer's suggestion we improved Figure 2 (see also response to comment R2.5). At Page 5 (lines 21 and 26), we clarified that the results from existing operational forecasting systems are stored into the Resource Layer and made available to the multy-model ensemble system. TMES outputs are also stored in the Resource Layer.*

**R2.13** Page 7 - Line 6-7: "hazard maps ... to identify vulnerable areas" there is a ix of the two terms that is common but not a good practice. I suggest following UNISDR Terminology where hazard is related with the physical aspects of the storm the coastal area and vulnerability with the socioeconomic aspects.

*Response: We concur with the reviewer that the hazard and vulnerability terms were not properly used. We modified the manuscript in the Introduction, section 2.3 and section 3.3 to clarify the terminology adopted in this study.*

**R2.14** Page 7 - Line 25: "It must be pointed out that the widely used Stockdon's..." The Stockdon formula is not applicable in rocky and gravel beaches. The problem is not the underestimation of the runup is the use of an inappropriate formula that results in underestimation of the runup.

*Response: The Stockdon's formula was improperly applied to all coastal segments. $R_2$ results presented in section 2.5 (Fig. 7) are now reported only for the coastal segments characterized by sandy beaches (see also response to comment R2.5).*

**R2.15** Figure 3: Substitute the "C" by "L" for low pressure.

*Response: Done.*

**R2.16** Page 12 - Line 11-13: There is a large number of local names that are not shown in the figure and is difficult to follow by the reader. Please add a more detail figure.

*Response: We removed the local names from the manuscript and simplified the sentence, that now reads as (Page 13, lines 3-4): "As a consequence, the sea flooded several coastal locations, where the firemen set up anti-flooding barrages."*

**R2.17** Page 14 - Line 27: "adaptation capacity" the dune and berm characteristics of a beach are not the adaptation capacity. The term adaptation is related with the ability of the system to overcome long term changes in forcing factors. Beach and berm characteristics can be combined with physical parameters (e.g. wave height water level) to calculate process based indexes that can serve as hazard intensity and extend parameters. A review of such indexes can be found in Ferreira, Ó., Plomaritis, T.A., Costas, S., 2017. Process based indicators to assess storm induced coastal hazards. Earth-Science Reviews 173, 159-167.

*Response: We concur with the reviewer that there was a mistake in the mentioned sentence. We modified the sentence as follow (Page 14, lines 35, Page 15, lines 1-2): "... comparing the magnitude of the impact (wave run-up for inundation and beach/shoreline retreat for erosion) with the morphological characteristics of the system (dune/berm height for inundation and beach width for erosion)."*

*We thank the reviewer for bringing this useful citation to our attention. We cited the study of Ferreira et al. (2017) in the Introduction (Page 2, line 19) and the discussion (Page 14, line 34).*

**References**

[revised manuscript text omitted]